# Modulation-doping a correlated electron insulator

Debasish Mondal [1], Smruti Rekha Mahapatra[1], Abigail M. Derrico[2], Rajeev Kumar Rai [3], Jay R. Paudel[2], Christoph Schlueter[4], Andrei Gloskovskii[4], Rajdeep Banerjee[1], Atsushi Hariki[5], Frank M. F. DeGroot [6], D. D. Sarma [1], Awadhesh Narayan[1], Pavan Nukala [3], Alexander X. Gray [2] ✉ & Naga Phani B. Aetukuri [1] ✉

Correlated electron materials (CEMs) host a rich variety of condensed matter phases. Vanadium dioxide ($VO_2$) is a prototypical CEM with a temperature-dependent metal-to-insulator (MIT) transition with a concomitant crystal symmetry change. External control of MIT in $VO_2$—especially without inducing structural changes—has been a long-standing challenge. In this work, we design and synthesize modulation-doped $VO_2$-based thin film heterostructures that closely emulate a textbook example of filling control in a correlated electron insulator. Using a combination of charge transport, hard X-ray photoelectron spectroscopy, and structural characterization, we show that the insulating state can be doped to achieve carrier densities greater than $5 \times 10^{21}$ cm$^{-3}$ without inducing any measurable structural changes. We find that the MIT temperature ($T_{MIT}$) continuously decreases with increasing carrier concentration. Remarkably, the insulating state is robust even at doping concentrations as high as ~0.2 e$^-$/vanadium. Finally, our work reveals modulation-doping as a viable method for electronic control of phase transitions in correlated electron oxides with the potential for use in future devices based on electric-field controlled phase transitions.

Strong electron-electron correlations within narrow d- or f-orbitals underpin a variety of condensed matter phenomena, such as metal-to-insulator transitions (MITs), high-temperature superconductivity, magnetism, and multiferroicity, often observed in correlated electron materials (CEMs)[1,2]. $VO_2$ is a prototypical example of a CEM with a temperature-dependent metal-to-insulator phase transition. The electronic phase transition in bulk $VO_2$, which occurs at an MIT temperature ($T_{MIT}$) of ~340 K, is accompanied by a structural phase transition from a metallic rutile phase to an insulating monoclinic phase[3,4].

The origin of the MIT in $VO_2$—whether gap-opening is driven by the symmetry-lowering structural transition or by electron-electron correlations—has been widely studied[5–8]. In particular, there is widespread interest in the nature of the insulating state and its external control via doping[9,10], strain[11], oxygen vacancy creation[12], hydrogenation[13], light-and-pulse-induced modulation[14,15], and via electric-fields in a field-effect transistor geometry[16–18].

For example, n-type doping of $VO_2$ with dopants such as $W^{6+}$, $Mo^{5+}$, and $Nb^{5+}$ was shown to decrease $T_{MIT}$ and stabilize the metallic phase[9,10,19]. By contrast, p-type doping of $VO_2$ with dopants such as $Cr^{3+}$, $Ga^{3+}$, and $Al^{3+}$ was shown to increase $T_{MIT}$, thereby stabilizing the insulating phase[20–22]. Similarly, both oxygen vacancy creation and hydrogenation were shown to n-dope $VO_2$ and decrease $T_{MIT}$[12,13,23].

[1]Solid State and Structural Chemistry Unit, Indian Institute of Science, Bengaluru, Karnataka, India. [2]Department of Physics, Temple University, Philadelphia, PA, USA. [3]Centre for Nano Science and Engineering, Indian Institute of Science, Bangalore, Karnataka, India. [4]Deutsches Elektronen-Synchrotron, DESY Hamburg, Germany. [5]Department of Physics and Electronics, Graduate School of Engineering, Osaka Metropolitan University, Osaka, Japan. [6]Utrecht University, Inorganic Chemistry and Catalysis Group Universiteitsweg 99, Utrecht, The Netherlands. ✉e-mail: axgray@temple.edu; phani@iisc.ac.in

Finally, in $VO_2$ thin films, macroscopic tensile strain along the rutile a-axis was also shown to decrease $T_{MIT}$[11,12].

In all these previous approaches, modulation of $T_{MIT}$ was always associated with macroscopic changes to the lattice parameters (due to strain) and/or dopant-induced local structural distortions[9,10,12,13,20,23]. In such experiments, where both the lattice strain and carrier concentration change, it is challenging and, sometimes, impossible to disentangle the role of carrier concentration changes from the role of lattice strain. For instance, in the case of W-doped $VO_2$, an increase in W-doping concentration increases both the carrier density and lattice strain[24].

Other techniques utilizing external stimuli, such as electric-field-induced metallization of $VO_2$ in a field-effect transistor geometry, could, in theory, enable the modulation of its conductivity without inducing macroscopic structural changes. However, previous attempts at electric-field-driven metallization of $VO_2$ were not successful[12,16,17,25,26]. For example, attempts at modulating the MIT in $VO_2$ in a field effect transistor geometry yielded less than a 1 K change in $T_{MIT}$[27,28]. The weak response of $T_{MIT}$ of $VO_2$ to external electric field, even when gated through high-K dielectrics, was attributed to the presence of strong correlations in the insulating $VO_2$ phase[16]. Further, ionic-liquid gating of $VO_2$, which could enable accessibility to large interfacial electric-fields, led to oxygen vacancy creation and/or hydrogenation of $VO_2$[12,16]. Heterostructures with differing compositions such as for example $VO_2/W_xV_{1-x}O_2$ based heterostructure thin films showed a larger change in $T_{MIT}$[29]. However, these changes are related to elemental doping driven by a chemical potential mismatch of the dopant-ion ($W^{6+}$ in this specific case). By stark contrast, we propose modulation-doping of $VO_2$ using electronic chemical potential differences at oxide heterostructures.

Modulation- or remote-doping of oxide semiconductors is an alternative method for achieving high dopant carrier densities without inducing local structural distortions[30–33]. In modulation-doping, a chemical potential mismatch between a high band gap heavily doped layer (dopant-layer) and a lower band gap undoped layer (channel) leads to charge transfer from the heavily doped dopant-layer to the undoped channel. In general, the dopant layer and the channel are spatially separated by a barrier (or a spacer) layer that kinetically limits the interdiffusion of the dopants from the dopant layer to the channel layer while allowing charge transfer via quantum mechanical tunneling[34,35].

Modulation-doping was successfully applied to semiconductors and band-insulating oxides such as ZnO and $SrTiO_3$[31,33,36–39]. However, modulation-doping of correlated electron insulators has had limited success. For example, Stemmer and colleagues reported modulation-doping of $NdNiO_3$, but this did not lead to any significant changes in $T_{MIT}$ of the nickelate[31]. Whether modulation-doping could be a generic approach to induce phase transitions in oxides is unclear and several key questions remain unanswered. For example, how do bands evolve in correlated oxides as a function of doping? Can a rigid band model be applied to understand doping in oxides? How sensitive are the ground state properties in correlated oxides to carrier doping?

In this work, we address some of these open questions using the MIT in $VO_2$ as a model system. We propose a modulation-doped heterostructure to n-dope $VO_2$ without inducing any structural distortions. Commonly in modulation-doping, an epitaxial structure is grown with a spacer, and the dopant layers are epitaxially matched to the semiconducting channel layer. Note that both the spacer and dopant layers must be insulating with a bandgap that is higher than that of the channel layer. However, the only stable rutile oxide that is insulating with a compatible band mismatch that allows modulation-doping of $VO_2$ is $TiO_2$. The other rutile oxides such as $CrO_2$, $RuO_2$, and $IrO_2$ are metallic and therefore not compatible as dopant layers[40,41].

As an additional challenge, oxygen lattice continuity in epitaxial structures might also lead to oxygen vacancy diffusion across the layers[23,42,43]. We note that oxygen vacancy formation, which was shown to affect the MIT in $VO_2$, is commonly observed in transition metal oxides[12,44,45]. Thus, in order to prevent oxygen vacancy diffusion across the wide band-gap spacer layer as well as to circumvent the lack of lattice-matching insulating rutile oxides, we have gone away from epitaxially-matched modulation-doped heterostructures. Instead, we propose an amorphous $LaAlO_3$ (LAO) layer (with a reported electronic band gap of ~5.6 eV)[46] as the spacer layer. Since LAO has a low oxygen vacancy-diffusivity[47], we use an amorphous oxygen-deficient $TiO_{2-x}$ as the dopant layer. Stoichiometric $TiO_2$ has a bandgap of ~3 eV[48] and $TiO_{2-x}$ is n-type conducting. Using $TiO_{2-x}$ instead of a conventionally doped $TiO_2$ such as Nb-doped $TiO_2$, significantly simplified heterostructure deposition. Furthermore, this approach avoids the interdiffusion of metallic dopants such as Nb in Nb-doped $TiO_2$ and the associated unintentional doping of $VO_2$.

The modulation-doped structure for all samples used in this work is comprised of a $VO_2$ channel layer, a 2 nm thick LAO spacer layer, and a 3 nm thick $TiO_{2-x}$ dopant layer, as shown schematically in Fig. 1a. All heterostructures were capped with a 1 nm thick LAO layer to prevent dopant passivation from atmospheric impurities as well as oxidation of the $TiO_{2-x}$ dopant layer. Fermi level alignment across the structure is expected to lead to an electron accumulation region at the $LAO/VO_2$ interface. Expected band-alignments for this type-I heterojunction before and after heterostructure formation are shown in Fig. 1b.

## Results

To experimentally realize the proposed modulation-doped structure, all samples were grown using pulsed laser deposition (PLD) on single-crystalline $TiO_2$ (001) substrates. $VO_2$ was deposited at 425 °C, while all the other amorphous layers were deposited at room temperature (see methods section for details). It is important to note that room-temperature deposition of the spacer, dopant and capping layers also minimizes any interfacial interdiffusion. A cross-sectional scanning transmission electron microscopy (STEM) image (Fig. 1c) and the associated energy dispersive spectroscopy (EDS) elemental maps (Fig. 1d) show abrupt high-quality interfaces between the $TiO_2$ substrate and the $VO_2$ film, and between the film and LAO spacer. Furthermore, the STEM image of pristine $VO_2$ on $TiO_2$ (001) substrate shows (supplementary Fig. 1) that the epitaxial atomic arrangement of $VO_2$ is identical for $VO_2$ heterostructures and thin films. This data is consistent with in-situ RHEED patterns of the deposited $VO_2$ films and suggests that the films are both atomically smooth and single-crystalline (Supplementary Fig. 2). The atomic force microscopy (AFM) images of the complete heterostructures further confirm the high quality of the growth by showing atomically smooth film surfaces (Supplementary Fig. 3).

To study the correlation between modulation-doping-induced carrier density changes and the changes in the MIT characteristics, we deposited several modulation-doped heterostructures with varying thicknesses of the $VO_2$ layer ranging from 1.5 nm to 9.5 nm, while keeping the thickness of the $TiO_{2-x}$ layer unchanged at 3 nm. Thomas-Fermi screening lengths in $VO_2$ are expected to be on the order of 1 nm (Supplementary Note 1) and, therefore, the highest n-type carrier densities are expected for the lowest $VO_2$ film thickness used in this study (~1.5 nm). Heterostructures with $VO_2$ films thinner than ~1.5 nm were not attempted due to the expected titanium interdiffusion at the $VO_2$ film and $TiO_2$ substrate interface[49,50]. We note that interfacial titanium interdiffusion will be present in thicker films as well. However, at $VO_2$ thicknesses greater than 1.5 nm, there is still an observable MIT.

A summary of the θ−2θ X-ray diffraction (XRD) measurements, performed at room temperature, for a 9.5 nm $VO_2$ film and the $VO_2/LAO/TiO_{2-x}/LAO$ heterostructures on $TiO_2(001)$ substrates are shown in Fig. 2a. Clearly, the angular positions of the $VO_2(002)$ Bragg reflection peaks are identical for both the 9.5 nm $VO_2$ film (purple spectrum) and the 9.5 nm $VO_2$ heterostructure (blue spectrum). Furthermore, it is

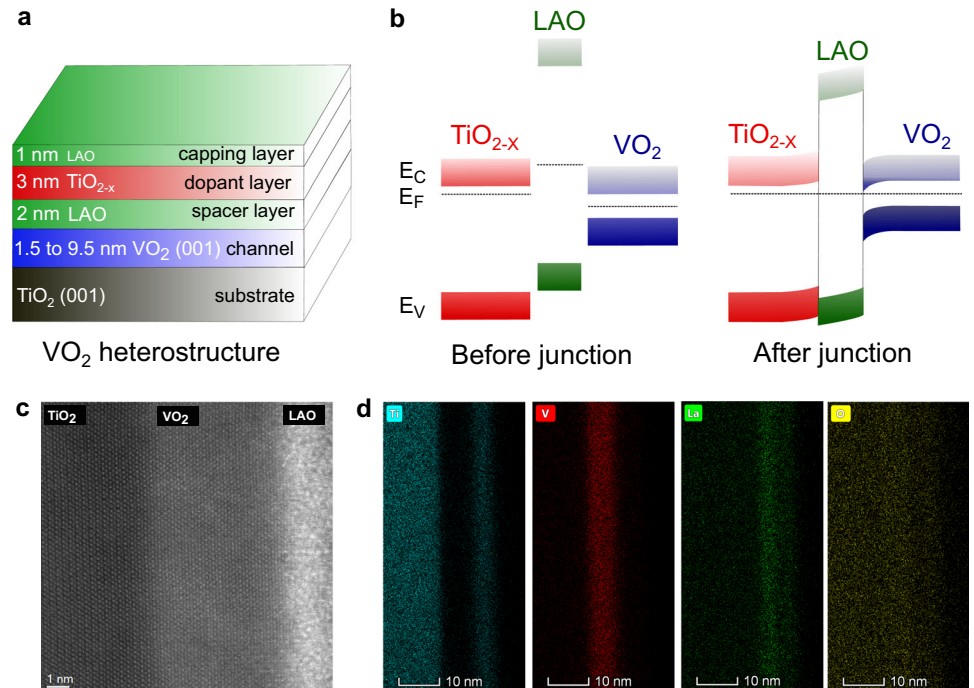

**Fig. 1 | Structure of Modulation-doped VO$_2$ heterostructures. a** Schematic diagram of the heterostructures used in this work. The thickness of VO$_2$ is varied while the thicknesses of all the other layers are as mentioned in the schematic. **b** Schematic energy band diagram for a VO$_2$/LAO/TiO$_{2-x}$ heterostructure before and after the junction formation. Electron accumulation is expected based on the known band offsets. The color intensities are chosen to be proportional to expected electron densities for better visualization. E$_C$, E$_V$, and E$_F$ indicate the conduction band edge, valence band edge, and Fermi level, respectively. **c** High-resolution cross-sectional high-angle annular dark-field scanning transmission electron microscopy (HAADF-STEM) image showing abrupt interfaces between TiO$_2$ substrate and VO$_2$ film and VO$_2$ film and the amorphous LAO spacer layer. **d** Elemental mapping using energy dispersive x-ray spectroscopy (EDS) showing the various layers in the heterostructure. Note that the scales of **c** and **d** are different.

evident that the angular position of the Bragg reflection is essentially independent of the thickness of the VO$_2$ film in the heterostructure. Additionally, there were no significant changes in the θ-2θ X-ray diffractograms between VO$_2$ films and heterostructures with the same VO$_2$ thickness (Supplementary Fig. 4). Reciprocal space maps also confirm that all samples are coherently strained in the plane of the TiO$_2$(001) substrate. The out-of-plane rutile c-axis lattice parameter is identical for the thin film and the heterostructures for all thicknesses of VO$_2$ (Supplementary Fig. 5). Based on θ-2θ XRD measurements, reciprocal space maps, and cross-sectional STEM imaging we conclude that there are no changes in the lattice parameters between VO$_2$ films and heterostructures. And therefore, changes in strain cannot account for the reduction in T$_{MIT}$ observed in modulation-doped VO$_2$ heterostructures (see Supplementary Figs. 1 and 4 and Supplementary Table 1). We also note that the reflections for the LAO spacer and capping layers and the TiO$_{2-x}$ dopant layer are absent, suggesting that these layers are not crystalline.

Next, we discuss the variations in the MIT characteristics for the same set of samples as used in the XRD studies. As shown in Fig. 2b, T$_{MIT}$ systematically decreases with decreasing VO$_2$ thickness in the heterostructure. Note that the decrease in T$_{MIT}$ for thin films of VO$_2$ is thickness-independent (Supplementary Fig. 6), suggesting that the decrease in T$_{MIT}$ for VO$_2$ heterostructures is intrinsic to heterostructure formation. Furthermore, the sheet resistance of VO$_2$ heterostructures in the insulating state also decreases (Supplementary Fig. 7). Except for the heterostructure with the thinnest VO$_2$ layer (1.5 nm), all films continue to show a positive temperature coefficient of resistance, suggesting metallicity above T$_{MIT}$.

We summarize our observations and compare the changes in T$_{MIT}$ with other previously published studies in Fig. 2c (also see Supplementary Fig. 8 for details regarding T$_{MIT}$ calculation). Significantly, there is a nearly 60 K change in T$_{MIT}$ for the thinnest

heterostructures without any measurable changes to the rutile C-axis. In contrast, any comparable change in T$_{MIT}$ in the literature is associated with ΔC$_R$ greater than 0.5 pm. This demonstrates control over the MIT in VO$_2$ without any measurable structural changes in the VO$_2$ heterostructures proposed and synthesized in this work. We note that a decrease in T$_{MIT}$ was observed for elemental doping of VO$_2$ with n-type dopants such as W and Mo, while an increase in T$_{MIT}$ was observed for hole-doping with elements such as Cr and Al. There is no W or Mo in any of the heterostructures in this work, and both La and/or Al doping can be ruled out because they would (if anything) lead to hole-doping resulting in an increase in the T$_{MIT}$. This is contrary to the decrease in T$_{MIT}$ observed in these VO$_2$ heterostructures.

To measure the extent and type of doping, we performed temperature-dependent Hall measurements. Hall measurements show an enhancement in the carrier densities in the insulating state with the Hall coefficient indicative of electron-doping (Fig. 3a). On the other hand, carrier densities in the insulating phase increased from ~6 × 10$^{17}$ cm$^{-3}$ (for 9.5 nm VO$_2$ thin film) to ~2.8 × 10$^{19}$ cm$^{-3}$ (for 9.5 nm VO$_2$ heterostructure) to a highest of ~5 × 10$^{21}$ cm$^{-3}$ for the 2.5 nm VO$_2$ heterostructure. Contrastingly, the metallic state carrier densities remain identical (~6 × 10$^{22}$ cm$^{-3}$) across all the VO$_2$ heterostructures and are consistent with previous reports[12,25].

However, carrier mobility decreases in both the insulating and metallic states as the thickness of the VO$_2$ layer decreases (Fig. 3b). In the metallic state, this is potentially due to contributions from interfacial scattering, which increases with decreasing film thickness. In the insulating state, the decrease in carrier mobility could result in part from increased electron-electron scattering and interfacial scattering. A summary of the changes in carrier concentration is plotted against T$_{MIT}$ in Fig. 3c. There is a clear correlation between the T$_{MIT}$ and the

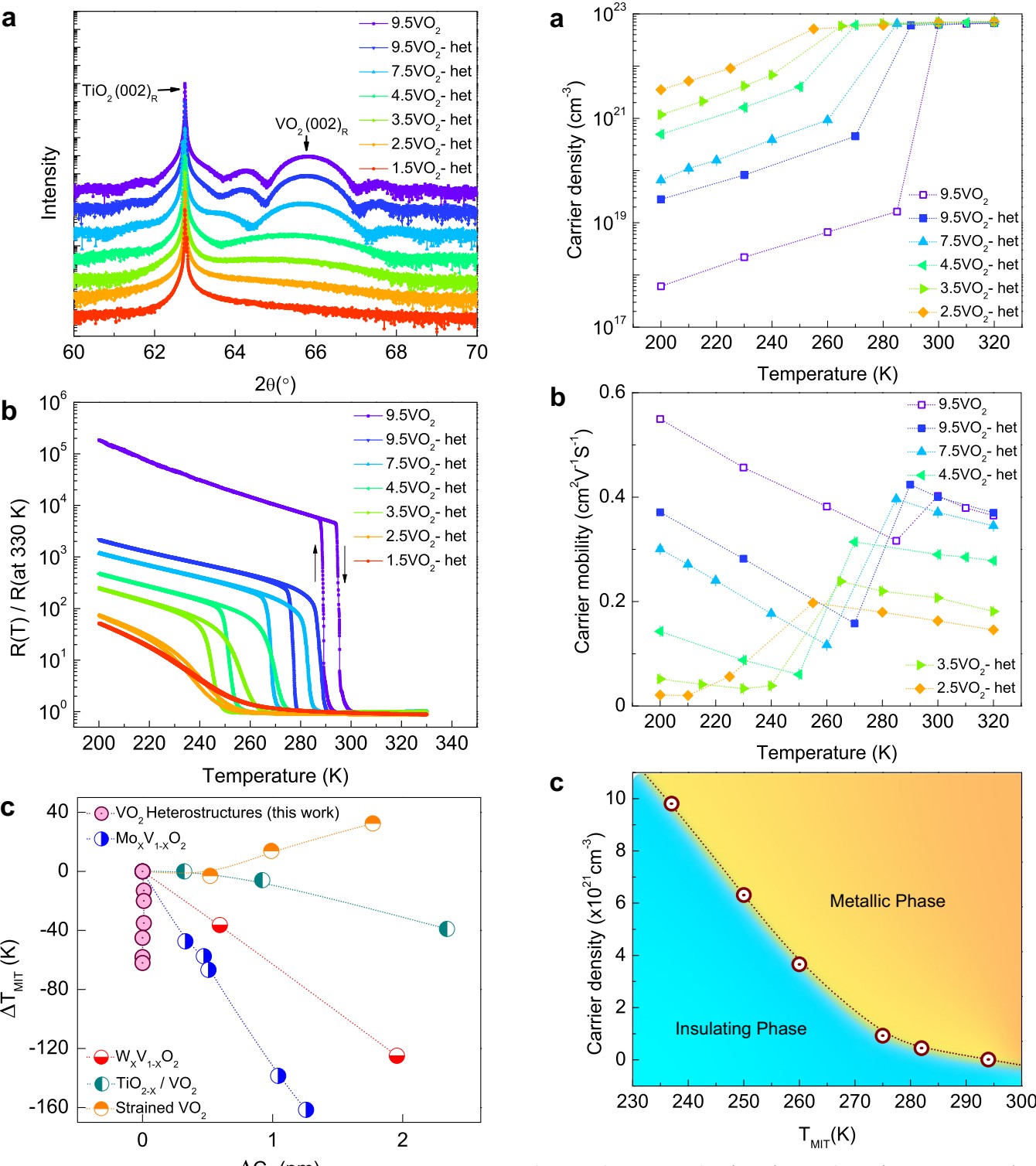

**Fig. 2 | XRD and temperature-dependent electrical transport in VO₂ heterostructures.** **a** High-resolution θ-2θ XRD spectra for 9.5 nm VO₂ thin film and for VO₂ heterostructures with varying thicknesses. For nomenclature simplicity, we distinguish VO₂ thin films and heterostructures with a VO₂ thickness of 't' as tVO₂ and tVO₂-het, respectively. **b** Resistance versus temperature plots for the same set of samples as shown in **a**. Resistance values presented here are normalized to the resistance at 330 K (also see Supplementary Fig. 7). **c** A comparison of the changes in $T_{MIT}$ versus the changes in the rutile C-axis lattice parameter ($\Delta T_{MIT}$ vs $\Delta C_R$) for this work and other previously published work using W- and Mo-doping[9,10], oxygen vacancy doping[22] and strain[11]. The relative changes are compared to the undoped and unstrained states in the case of bulk doping and for strained VO₂, respectively. [also see Supplementary Table 1].

**Fig. 3 | Carrier concentration-dependent MIT in VO₂ heterostructures.** Plots of temperature-dependent **a** carrier densities and **b** carrier mobilities for tVO₂ and tVO₂-het samples as mentioned in the legends. Carrier density in the insulating state increases with decreasing VO₂ thickness while carrier mobility decreases. **c** A phase diagram from the results in **a**. The dotted lines connected across the data points are a guide to the eye.

carrier density with the highest carrier density of ~5 × 10²¹ cm⁻³ stabilizing the metallic state of VO₂ to a $T_{MIT}$ of ~237 K (Supplementary Fig. 8). Importantly, the continuous increase in carrier density with decreasing VO₂ thickness without any lattice parameter changes is suggestive of modulation-doping in VO₂ heterostructures.

To further establish that most of the carriers are induced by modulation doping, we prepared two additional control samples. The first is a 7.5 nm $VO_2$ thin film with a 2 nm LAO cap layer (7.5$VO_2$-LAO) and the second is a 7.5 nm $VO_2$ heterostructure with a 3 nm stoichiometric $TiO_2$ layer (7.5$VO_2$-LAO-$TiO_2$). We compared the MIT characteristics of these two samples with the MIT characteristics of the 7.5 nm $VO_2$ thin film (7.5$VO_2$) and a 7.5 nm $VO_2$ heterostructure with a $TiO_{2-x}$ dopant layer (7.5$VO_2$-het). A summary of sheet resistance versus temperature data is shown in Supplementary Fig. 9. The 7.5 nm $VO_2$ heterostructure with a $TiO_{2-x}$ dopant layer has the lowest sheet resistance and the lowest $T_{MIT}$ with a $T_{MIT}$ change of ~20 K. By contrast, the decrease in $T_{MIT}$ was restricted to ~8 K after the deposition of the 2 nm LAO layer. Remarkably, there is no further decrease in $T_{MIT}$ in the heterostructure with a stoichiometric $TiO_2$ layer, suggesting that the $TiO_{2-x}$ dopant layer is required for the observed large change in $T_{MIT}$.

Consistent with this, the carrier density in the 7.5 nm $VO_2$ heterostructure with the $TiO_{2-x}$ dopant layer is ~7 × $10^{19}$ cm$^{-3}$ compared to a carrier density of 2 × $10^{19}$ cm$^{-3}$ for the 7.5 nm $VO_2$ capped with 2 nm LAO (Supplementary Fig. 10). It is possible that amorphous (disordered) LAO could host ionized donors and enable modulation-doping[33]. However, we found that the amorphous LAO deposited for these experiments is insulating, suggesting that any ionized donors should be below the measurement threshold of electrical resistivity measurements. We estimate that such ionized donors in LAO, if any, should have a carrier density of ~$10^{19}$ cm$^{-3}$ (assuming a carrier mobility of 0.01 cm$^2$/V·s) or lower, putting an upper bound on the number of carriers contributed by the spacer layer.

As further proof of modulation-doping, we performed resistance-temperature measurements on heterostructures with varying LAO spacer layer thicknesses of 2 nm, 4 nm, and 10 nm. The thickness of $VO_2$ is fixed at 7.5 nm and that of $TiO_{2-x}$ dopant layer is fixed at 3 nm for all three heterostructures. As the thickness of the LAO layer ($t_{LAO}$) increases, the probability of charge transfer from the $TiO_{2-x}$ dopant layer to the $VO_2$ layer decreases (Fig. 4a, b). Consistent with this, at the highest $t_{LAO}$ of 10 nm, where the lowest amount of charge transfer is expected from the dopant layer, $T_{MIT}$ is the closest to that of a 7.5 nm $VO_2$ film with a 2 nm LAO cap layer, but without the dopant layer. Clearly, the dopant layer does not significantly affect the transition temperature when a 10 nm thick LAO spacer layer is used (Fig. 4c). By contrast for the heterostructure with $t_{LAO}$ = 2 nm, the $T_{MIT}$ is shifted by ~20 K as discussed earlier. Since the thicknesses of the other layers are

fixed, the larger shift in $T_{MIT}$ for the $t_{LAO}$ = 2 nm heterostructure implies an increased charge transfer for the thinner spacer layers. This further reinforces the central conclusion that the shift in the transition temperature is enabled by modulation doping.

To probe the electronic band bending that enables electron accumulation in the $VO_2$ channel layer, we performed bulk-sensitive hard X-ray photoelectron spectroscopy (HAXPES)[51] measurements at the P22 beamline in the PETRA III synchrotron at DESY. We note that standard ultra-violet photoemission (UPS) or soft X-ray photoemission measurements cannot facilitate a probing depth sufficient to reach the $VO_2$/LAO interface that is buried beneath multiple layers of the heterostructure. To capture interfacial band bending in $VO_2$, HAXPES measurements were performed in both the insulating phase (at 200 K) and the metallic phase (at 310 K) for $VO_2$ heterostructures with $VO_2$ thicknesses of 1.5 nm, 3.5 nm, 4.5 nm, and 7.5 nm, and for a $VO_2$ thin film with a thickness of 7.5 nm as a reference.

For the 7.5 nm $VO_2$ film measured at 200 K (insulating state), the binding energies of the V $2p_{3/2}$ and V $2p_{1/2}$ core-level peaks were observed to be ~515.8 eV and ~523.1 eV, respectively (see Fig. 5a). These measured binding energies (see Supplementary Fig. 11 for binding energy calibration procedure) are consistent with previous reports[52–54]. Importantly, a systematic shift of the main component of the V $2p_{3/2}$ peak to higher binding energies is observed for the $VO_2$ heterostructures, with the highest increase in binding energy (~250 meV) observed for the heterostructure with the thinnest $VO_2$ layer (1.5 nm), as seen in the inset. This is also in agreement with the highest carrier densities and the lowest $T_{MIT}$ being observed for the heterostructures with the thinnest $VO_2$ layers. For measurements performed on $VO_2$ in the metallic state, no such binding energy shift was observed (Fig. 5b and Supplementary Fig. 12). This is consistent with the complete screening of interfacial electric fields at the metallic $VO_2$/LAO interface (Fig. 5c, d). The presence of binding energy shifts observed in the insulating state of $VO_2$ and their absence in the metallic state of $VO_2$ further support carrier doping by chemical potential shifts (modulation-doping) in the insulating state for $VO_2$ heterostructures as proposed in this work.

Photoemission data also showed two remarkable features in the V $2p$ spectra. The first, labeled *P1* in Fig. 5a, b, is a lower binding energy shoulder around ~514.5 eV in the insulating and metallic states. The presence of this spectral feature at lower binding energies was proposed to signify non-local screening from coherent $3d^1$ states near $E_F$ in the metallic phase of $VO_2$[54,55]. Interestingly, the intensity of *P1* in

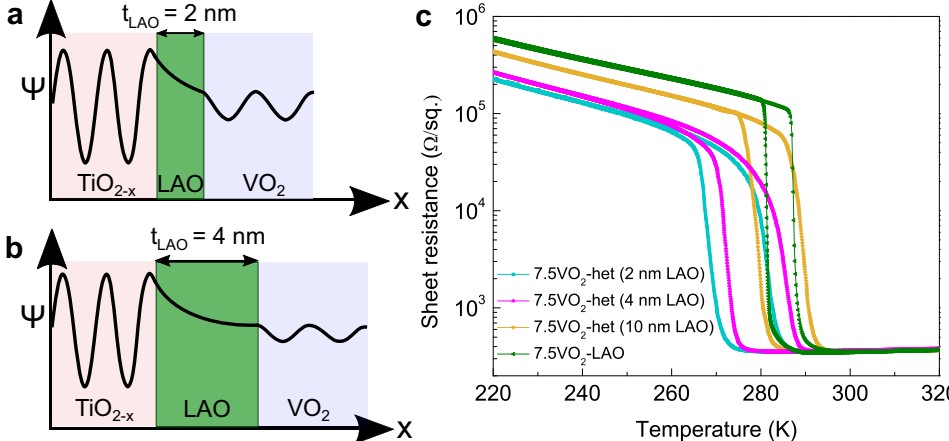

**Fig. 4 | Transport characteristics of heterostructures as a function of spacer layer thickness.** Schematic representation of quantum mechanical tunneling of charge carrier across the spacer layer for **a** 2 nm and **b** 4 nm thicknesses of LAO ($t_{LAO}$). '$\Psi$' is the electronic wave function. The schematics represent a decrease in the transfer of charge carriers from the dopant layer ($TiO_{2-x}$ layer) to the $VO_2$ layer

as a function of increasing $t_{LAO}$. **c** Resistance-Temperature plots comparing the MIT characteristics of 7.5 nm $VO_2$ modulation-doped heterostructures employing $t_{LAO}$ of 2 nm, 4 nm, and 10 nm with the MIT characteristics of a 7.5 nm $VO_2$ film with a 2 nm LAO cap layer, but without any dopant layer.

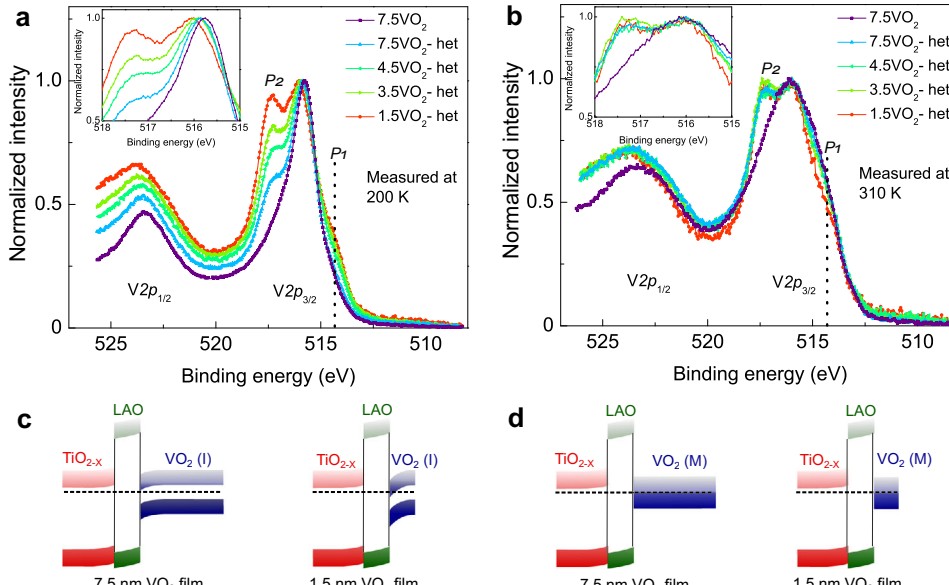

**Fig. 5 | Band bending and core level spectral changes in VO₂ heterostructures.** A comparison of V $2p$ core-level spectra of modulation-doped VO₂ heterostructures for **a** the insulating (200 K) and **b** the metallic states (at 310 K). A clear shift in the V $2p$ levels is seen in the insulating state spectra but not in the metallic state spectra. Schematics in **c** and **d** show the expected band-bending in the modulation-doped heterostructures for the insulating and metallic states respectively. Band-bending is expected in the insulating state and not in the metallic state. Two additional spectral features not seen in VO₂ thin films are labeled *P1* and *P2*.

the insulating state increases with increasing carrier density and decreasing VO₂ thickness in the heterostructures. The emergence of this peak in the insulating state spectra for VO₂ heterostructures suggests that the additional charge transferred to the VO₂ channel layer enables non-local screening that was previously observed only in the metallic phase of VO₂.

To further quantify the evolution of the *P1* peak across the metallic and insulating phases, we compared the metallic and insulating state spectra of 1.5 nm and 7.5 nm VO₂ heterostructures in Supplementary Fig. 13. The intensity of the *P1* peak in the insulating state of the 7.5 nm VO₂ heterostructure is observable but small in comparison to the *P1* peak in the metallic state (Supplementary Fig. 13a). The intensity difference spectrum shows a large difference at ~514.5 eV (Supplementary Fig. 13c) further confirming that the non-locally screened shoulder is negligibly small in the insulating state when compared to the metallic state. Remarkably, the non-locally screened shoulder is quite predominant in the insulating state spectra of 1.5 nm VO₂ heterostructure (Supplementary Fig. 13b). The intensity difference spectra between the metallic and insulating states show a very small difference between the two phases (Supplementary Fig. 13d). These trends are consistent with the differences in the carrier densities between the metallic and insulating phases. The carrier density ratio between the metallic and insulating states is about 10 for the 1.5 nm VO₂ heterostructure compared to about 1000 for the 7.5 nm VO₂ heterostructure.

The second remarkable feature in the photoemission data is a high binding energy spectral feature at ~517.5 eV. This feature is associated with the V $2p_{3/2}$ peak and labeled *P2* in Fig. 5a, b. A corresponding feature is also observed for the V $2p_{1/2}$ peak but is more smeared out and appears as broadening on the higher-binding-energy side at ~525 eV. In general, higher binding energy spectral features are associated with higher oxidation states in photoemission. The presence of $V^{5+}$ is a possibility. However, an increase in the oxidation state from $V^{4+}$ to $V^{5+}$ cannot explain the observed electron-doping in VO₂ heterostructures. Generally, electron doping should decrease the $V^{4+}$ oxidation state in VO₂, and therefore, an increase in the oxidation state of vanadium cannot explain the increase in electron density in the VO₂

heterostructures. Therefore, $V^{5+}$, even if present, has no bearing on the MIT observed in heterostructures.

*P2* was also observed in VO₂ samples capped with 2 nm LAO (Supplementary Fig. 14). Therefore, we have also inspected the La $3d_{5/2}$ and Al $1s$ spectra to look for any chemical shifts associated with a redox or chemical reaction at the VO₂/LAO interface. As shown in Supplementary Fig. 15, there are no observable changes to the spectra across heterostructures. Finally, interfacial oxygen vacancy creation remains a possibility. However, any oxygen vacancy creation should lead to $V^{3+}$ and an associated low-binding-energy feature in both the metallic and insulating state spectra. However, the spectra do not show any signatures of oxygen vacancies in VO₂ heterostructures. Therefore, we rule out any oxygen-vacancy induced carrier doping in these heterostructures.

Furthermore, the intensity of *P2* is carrier density dependent. For all V $2p$ spectra in the metallic state, where the carrier density is independent of the VO₂ thickness, the intensity of this additional peak relative to the main V $2p_{3/2}$ peak is also independent of the VO₂ thickness, with the intensity ratio of *P2* to V $2p_{3/2}$ being close to 1. Contrastingly, the intensity of *P2* increases with the decreasing film thickness in the insulating state of VO₂. The intensity ratio of *P2* to V $2p_{3/2}$ approaches the intensity ratio observed for the metallic state spectra at the highest carrier density of ~5 × 10²¹ cm⁻³ in the insulating state. These carrier-density-dependent changes suggest that this new spectral feature is intrinsic to the heterostructure. However, this additional spectral feature might benefit from further spectroscopic investigation with complementary techniques such as X-ray absorption spectroscopy to confirm its origins.

Furthermore, since there is a strong correlation between electron density and *P2* peak intensity, we performed an LDA + DMFT Anderson impurity model calculation for the undoped and electron-doped VO₂ to examine the V $2p$ XPS spectral changes with electron doping. The calculation of the V $2p$ XPS spectrum shows that electron doping creates a satellite peak at approximately 517.5 eV (see Supplementary Note 2 and Supplementary Fig. 16 for details of the calculations). Since the V-O covalency for the $V^{3+}$ ($d^2$) state is weaker than $V^{4+}$ ($d^1$) one, the binding energy of $V^{3+}$ in electron-doped VO₂ increases due to a weaker

bonding and anti-bonding splitting in the XPS final states. Therefore, we assign the *P2* peak to a satellite induced by electron doping.

The combination of electron transport and HAXPES data show that VO$_2$ heterostructures facilitated effective modulation-doping and carrier densities as high as $5 \times 10^{21}$ cm$^{-3}$ could be achieved using this approach. The highest carrier densities correspond to electron doping of ~0.2 e$^-$/vanadium. This is an extremely high dopant density at which conventional rigid band models predict metallization in doped correlated insulators[56].

Bulk-sensitive valence-band HAXPES spectra recorded for the same set of heterostructures as discussed in Fig. 5 are shown in Fig. 6 and Supplementary Fig. 17. The insulating-state spectra for all VO$_2$ heterostructures (blue) exhibit nearly zero spectral intensity at the Fermi level while an appreciable non-zero spectral intensity is observed for the higher-temperature metallic-state spectra (orange). These spectra further confirm that VO$_2$ continues to undergo an MIT even in the presence of electron densities as high as ~0.2 e$^-$/vanadium. The presence of MIT at such high doping levels, without any observable changes in the lattice parameters (Fig. 2a and Supplementary Fig. 4), points to a possible renormalization of the electronic structure with doping.

What is unclear however is the presence or absence of the structural phase transition. To probe this, we performed temperature-dependent XRD measurements at 320 K (metallic phase) and 200 K (insulating phase) for a 9.5 nm VO$_2$ thin film and 9.5 nm and 4.5 nm VO$_2$ heterostructures. Figure 7a shows that there is a clear shift in the angular position of the out-of-plane Bragg reflection. This was attributed to the rutile to monoclinic phase transition in previous studies[11]. Similar changes to the out-of-plane Bragg reflection were also observed for the 9.5VO$_2$-het and 4.5VO$_2$-het across the MIT (Fig. 7b, c). Additionally, the X-ray diffractogram measured at 280 K for the 4.5VO$_2$-het shows rutile phase characteristics akin to that observed for the diffractogram measured at 320 K (Fig. 7d) suggesting that the electronic phase transition is concomitant with the structural phase transition.

Moreover, in Supplementary Fig. 18, we have also compared the XRD spectra of 9.5VO$_2$ and 9.5VO$_2$-het both in the insulating and metallic phases. It is clear from the figure that there is an excellent overlap of the diffractograms for the film and heterostructure, including thickness fringes, for both the structural phases. This is evidence that modulation-doping decreases the phase transition temperature but does not suppress the structural transition. The presence of the structural phase transition even at carrier densities as high as $5 \times 10^{21}$ cm$^{-3}$ is suggestive of a strong electron-lattice coupling in VO$_2$.

In summary, we demonstrated a purely electronic control of the MIT in modulation-doped VO$_2$ heterostructures. Our work shows that the insulating state in VO$_2$ is surprisingly robust even in the presence of electron doping as high as 0.2 e$^-$/vanadium. Notably, all the films meet the Mott criterion ($a_B \cdot n_C^{\frac{1}{3}} > 0.25$, where $a_B$ is the effective Bohr radius and $n_C$ is the carrier density). Therefore, metallicity is expected at all temperatures based on the carrier densities achieved in these experiments. We note that a similar robust insulating state had also been found in modulation-doped nickelate thin films[31]. Perhaps, the development of theoretical models that go beyond the conventional carrier concentration independent rigid-band models will be required to understand electronic phase transitions in correlated electron oxides. An alternate explanation could be that the insulating state is favored at lower thicknesses due to interfacial disorder-induced Anderson-like localization of carriers, which will be more pronounced for the thinnest films. Further experiments will be needed to assess whether the insulating state is stabilized by the interfacial disorder.

A remarkable feature of this work is the possibility of bulk metallization in modulation-doped VO$_2$. In general, in band semiconductors such as SrTiO$_3$, conductivity modulation is achieved over a thickness of 1–2 nm in the vicinity of the channel/spacer interface[33,57]. In this work, a sharp MIT is observed for heterostructures with VO$_2$ thicknesses as high as 9.5 nm, which are much higher than the Thomas Fermi screening length of ~1 nm. This is suggestive of the entirety of the film being metallized at the lowered T$_{MIT}$ after modulation-doping, even though the charge transfer densities are the highest at the interface (within 1–2 nm of the interface). While further studies are required to establish this beyond doubt, interfacial-doping induced

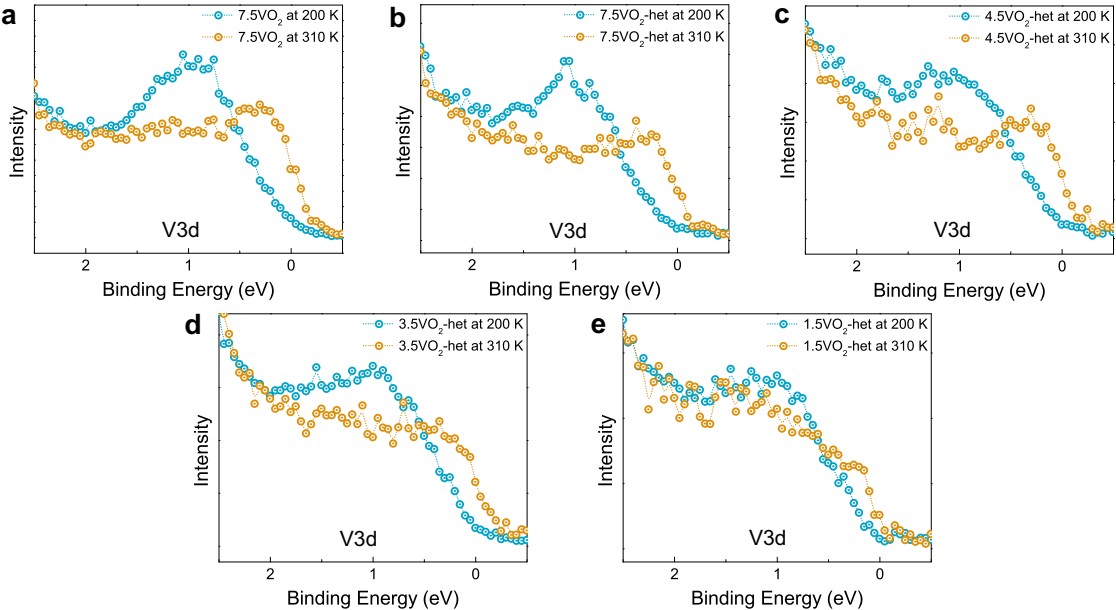

**Fig. 6 | Robust MIT in VO$_2$ at high carrier densities.** A comparison of the V 3d valence band (VB) spectra of modulation-doped VO$_2$ heterostructures for the insulating (200 K, blue) and metallic states (at 310 K, orange) for different VO$_2$ film thicknesses of **a** 7.5 nm VO$_2$ film and **b** 7.5 nm, **c** 4.5 nm, **d** 3.5 nm, and **e** 1.5 nm VO$_2$ heterostructures. There is a clear spectral weight shift across the MIT for all the samples with the insulating state being robust even for the heterostructure with a VO$_2$ thickness of 1.5 nm, corresponding to carrier doping of ~0.2 e$^-$/Vanadium.

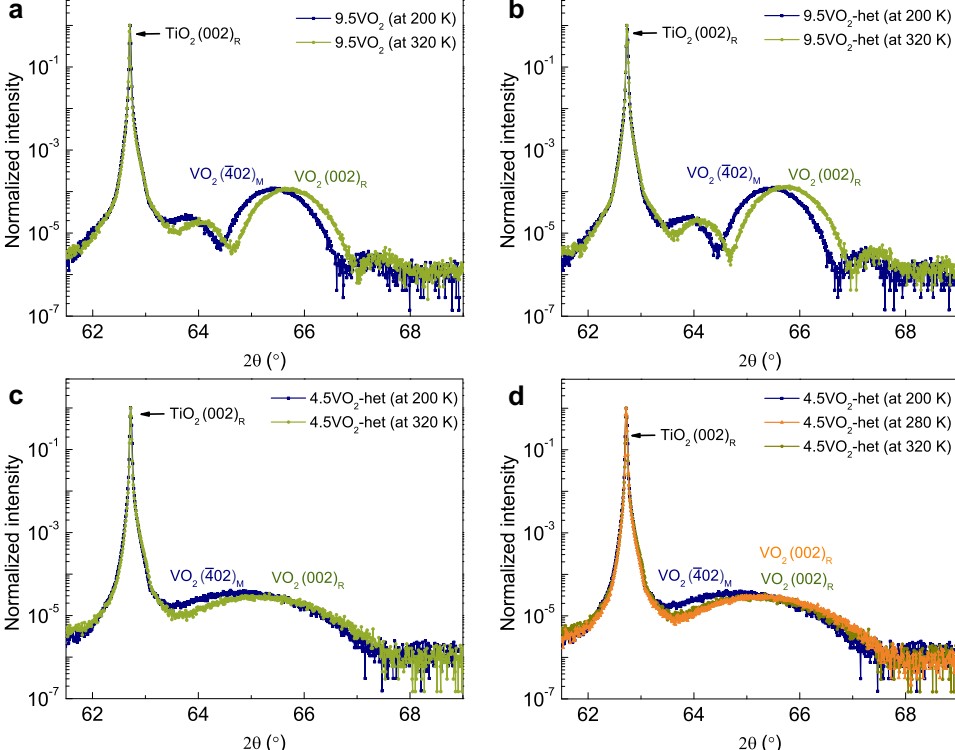

**Fig. 7 | Temperature-dependent X-ray diffractograms of VO$_2$ thin films and heterostructures.** X-ray diffractograms of **a** 9.5VO$_2$, **b** 9.5VO$_2$-het, and **c** 4.5VO$_2$-het measured in the insulating phase (at 200 K) and the metallic phase (at 320 K) of VO$_2$. VO$_2$ ($\overline{4}$02)$_M$ and (002)$_R$ reflections can be clearly distinguished for all the heterostructures studied in this work. The ($\overline{4}$02)$_M$ peak is the out-of-plane (of the substrate) Bragg reflection in the monoclinic phase of VO$_2$ while (002)$_R$ is the out-of-plane reflection in the rutile phase. **d** A comparison of temperature-dependent XRD spectra of the 4.5VO$_2$-het film measured at 200 K, 280 K, and 320 K. The XRD spectrum measured at 280 K shows rutile phase characteristics like the one measured at 320 K suggesting that VO$_2$ remains in the rutile phase at 280 K in the 4.5 nm VO$_2$ heterostructure.

bulk metallization of correlated electron insulators has implications for low-power electronics with high ON/OFF ratios[32,58].

Finally, our work shows that modulation-doping is a powerful technique for achieving high carrier densities, close to those possible with elemental doping. Since our approach does not need any epitaxially matched spacer and dopant layers, it expands the library of materials that can be explored for the study of modulation-doping-induced electronic phase transitions of other related CEMs including complex oxides[2] and pyrochlores[59]. This methodology, therefore, paves the way for exploring 'pure' electronic effects in correlated oxides and related systems. Such studies could also enable a fundamental understanding of band matching and relevant energy scales in complex oxides and, perhaps enable the discovery of new interfacial phases and devices that rely on phase transitions.

## Methods

Prior to deposition, single-crystalline rutile TiO$_2$ (001) substrates (Shinkosa, Japan) were treated using the procedure discussed previously[60]. All thin film samples were deposited using PLD (NEO-CERA) with a 248 nm KrF laser. All VO$_2$ thin films (both pristine films and heterostructures) were deposited on treated TiO$_2$ substrates from a sintered V$_2$O$_5$ target with a laser fluence of ~1.5 J/cm$^2$, 8 mTorr of oxygen pressure, and a growth rate of ~4.7 × 10$^{-2}$ Å/pulse at a substrate temperature of 425 °C[60]. For all heterostructure samples, 2 nm thick LAO spacer layers were deposited at 10 mTorr of O$_2$ pressure at a growth rate of ~5 × 10$^{-2}$ Å/pulse using a single-crystalline LAO target (Shinkosa Japan). 3 nm thick TiO$_{2-x}$ dopant layers were then deposited using a TiO$_{2-x}$ single-crystalline target (Shinkosa Japan) at a growth rate of ~4.2 × 10$^{-2}$ Å/pulse in 10$^{-5}$ Torr of background vacuum. Finally, a 1 nm thick LAO layer was deposited under the same conditions used for the

LAO spacer layer. Depositions of the spacer, dopant, and capping layers were all done at room temperature at a laser fluence of ~1.2 J/cm$^2$. For all depositions, the substrate-to-target distance and the laser pulse frequency were maintained at 55 mm and 2 Hz respectively.

High-resolution Cu-$K\alpha$ X-ray diffraction spectra for both the pristine and heterostructure VO$_2$ films were recorded in standard θ−2θ geometry using a Rigaku Smart Lab X-ray diffractometer. LEPTOS 7.8 software by Bruker was used to determine film thicknesses and to calculate the differential strain between pristine and heterostructure films.

Cross-sectional scanning transmission electron microscopy (STEM) imaging and energy dispersive x-ray spectroscopy (EDS) mapping were performed using TITAN Themis microscope (60–300 kV) equipped with a probe corrector and super-X four quadrant EDS detector. The high angle annual dark field (HAADF)-STEM images were acquired at an operating potential of 300 kV with a convergence angle of 24.5 mrad, 160 mm camera length, and a dwell time of 12 μs per pixel. The images were further processed with Gatan digital micrograph software. The EDS maps were acquired using Velox software under similar microscopic conditions with a dwell time of 2 μs per pixel.

Sheet resistance vs temperature measurements were performed in Van der Pauw geometry using Keithley 2450 SMU and Eurotherm 2408 PID temperature controller. Continuous temperature scanning was carried out at a rate of 4 K/min for both heating and cooling cycles.

To extract carrier density and mobility, Hall measurements for all films and heterostructures were performed using Van der Pauw geometry in a PPMS-Dynacool equipment from Quantum Design and Keithley SMU 2450 from Tektronix. For these measurements, the magnetic field was swept from 0 T to 2 T to −2 T to 0 T at a scan rate of 100 Oe/s for different temperatures ranging from 200 K to 320 K.

High-resolution hard X-ray photoelectron spectroscopy (HAXPES) measurements[61] were carried out with an incident photon energy of 6.2 keV at the sample temperatures of 200 K (insulating phase) and 310 K (metallic phase). Binding energy calibration was carried out using a high-resolution Fermi-edge measurement on a standard Au sample. Core-level and valence-band spectra were measured using a wide acceptance angle SPECS Phoibos 225HV hemispherical electrostatic analyzer in a near-normal emission geometry. The total energy resolution was estimated to be approximately 320 meV. Preliminary HAXPES measurements and sample screening were carried out using a lab-based HAXPES instrument at Temple University equipped with a wide acceptance angle ScientaOmicron EW4000 analyzer at a total experimental energy resolution of 450 meV.

To realize the origin of the *P2* peak (at 517.5 eV) in the V *2p* core level spectra (see Fig. 5), the simulations were performed on the electron-doped $VO_2$ system using LDA + DMFT Anderson impurity model calculations. Details of the calculations are presented in Supplementary Note 2 and Supplementary Fig. 16.

## Data availability
Data that support the findings of this study are available from the corresponding author upon reasonable request.

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

## Acknowledgements

A.X.G., A.M.D., and J.R.P. acknowledge support from the DOE, Office of Science, Office of Basic Energy Sciences, Materials Sciences, and Engineering Division under Award No. DE-SC0019297. The electrostatic photoelectron analyzer for the lab-based HAXPES measurements at Temple University was acquired through an Army Research Office DURIP grant (Grant No. W911NF-18-1-0251). We acknowledge DESY (Hamburg, Germany), a member of the Helmholtz Association HGF, for the provision of experimental facilities. Beamtime at DESY was allocated for proposal I-20210142. Funding for the HAXPES instrument at beamline P22 by the Federal Ministry of Education and Research (BMBF) under the framework program ErUM is gratefully acknowledged. A.X.G. also gratefully acknowledges the support from the Alexander von Humboldt Foundation. P.N. and R.K.R. acknowledge the Advanced Facility for Microscopy and Microanalysis (AFMM) for providing the electron microscope and FIB facility. A.N. acknowledges support from the startup grant at the Indian Institute of Science (SG/MHRD-19-0001). The authors acknowledge the micro nano characterization facility, national nanofabrication center, and the packaging lab at CeNSE, IISc for access to HR-XRD, wire bonding, and clean-room facilities. N.B.A. acknowledges the new faculty startup grant provided by the Indian Institute of Science under Grant No. 12-0205-0618-77. N.B.A. is thankful to Professor Anil Kumar for access to the PLD system. S.R.M. and D.M. want to thank Jibin J. Samuel and Mithun Ghosh for useful discussions. We thank Professor Satish Patil for providing access to facilities supported by the Swarnajayanti fellowship under Grant No. DST/SJF/CSA-01/2013-14. AFM measurements were performed on a Cypher-ES AFM funded by the DST-FIST program and Hall measurements were performed on a PPMS-Dynacool system funded under the UGC-CAS program. A.H. was supported by JSPS KAKENHI Grant Numbers 21K13884, 21H01003, 23H03816, 23H03817, and the 2023 Osaka Metropolitan University (OMU) Strategic Research Promotion Project for Younger Researcher. D.D.S. thanks Council of Scientific and Industrial Research for support.

## Author contributions

N.B.A. conceived the idea and designed the experiments and supervised the project. D.M. and S.R.M. deposited all the thin films and performed RHEED and XRD measurements. S.R.M. performed AFM measurements. D.M. performed electrical transport, Hall, RSM, and XRD measurements and data analysis. R.K.R. and P.N. performed cross-sectional STEM and EDS measurements. A.M.D., J.R.P., C.S., A.G., and A.X.G. performed HAXPES measurements. A.H. performed DMFT + LDA-Anderson calculations. F.M.F.D. contributed to HAXPES data analysis. R.B., D.D.S., and A.N. provided theoretical insights. N.B.A., A.X.G. planned and supervised HAXPES measurements; F.M.F.D. planned and supervised XPS calculations. N.B.A., D.M., and S.R.M. wrote the manuscript with contributions from A.X.G., P.N., and A.H. All authors reviewed and commented on the manuscript.

## Competing interests

The authors declare no competing interests.
