## [Peer Review File · Nature Communications]

REVIEWER COMMENTS

Reviewer #1 (Remarks to the Author):

This paper presents a systematic study combining the film synthesis, novel doping strategy and characterization (structure, composition and electrical) of VO₂ films. Given that VO₂ is one of the condensed matter systems that is heavily studied, it is perhaps worth noting that authors managed to discover a novel strategy to study electronic vs structural transition. It is further noteworthy (at least in this reviewer's opinion) that the use of amorphous layer of ultra wide bandage oxide LaAlO₃ as a space layer and amorphous oxygen deficient TiO₂ as a donor layer yields a controlled modulation doping strategy. As authors have pointed out this strategy eliminate the issue pertaining to lattice mismatch, and oxygen vacancy formation in VO₂ as well as eliminate the concerns related to transport owing to intermixing. Note that both La and Al on LaAlO₃ are not electron donors in VO₂.

Using a range of approaches combining transport, hard x-ray photoelectron spectroscopy, and structural characterization, authors demonstrated modulation doping of VO₂. High density of electron was achieved which allowed authors to control electronic transition (insulator to metal) without disturbing the structure. Results presented in Fig 2c is particularly noteworthy where a large change in T_{MIT} is achieved with no change in the lattice parameters.

The methodology using amorphous layer for modulation doping should also be relevant to other materials systems and hence, will likely be of broader interest.

Paper is very well written, and data is nicely presented. I recommend publication of the manuscript as is.

Reviewer #2 (Remarks to the Author):

The authors report on the effect of "modulation doping" on the metal-insulator transition (MIT) properties of VO₂ in the VO₂-based heterostructures. VO₂ is one of representative strongly-correlated oxides, showing the electronic phase transition (i.e., MIT) near room temperature. This

MIT is known to accompany the structural phase transition, and there has been a long-standing debate on the driving force of this phase transition, in particular on whether the electronic transition and the structural transition could be decoupled or not. To address this issue, the authors prepared the VO₂-based heterostructures with an insulating spacer layer made of LaAlO₃ and a carrier reservoir layer made of oxygen deficient TiO_{2-x}, and examined the impact of the charge-transfer doping from TiO_{2-x} to VO₂ on the MIT properties of VO₂. Then, based on the experimental results including XRD, STEM, transport, and HAXPES measurements, the authors conclude that they could modulate the MIT temperature (T_{MIT}) purely by electron doping without introducing structural deformation. They observed finite reduction of T_{MIT} together with the suppression of the resistance at the low-temperature insulating phase, both of which appeared to be associated with the electron doping as they in fact confirmed by the Hall-effect measurements. They also found that this effect became stronger as the thickness of the VO₂ layer was reduced. I agree that the electron doping in VO₂ should take place in their heterostructures, but I have a concern on their interpretation. I also have other concerns on the present study, which should preclude publication of this study, at least with the current version. The authors should address the following concerns in full before further consideration.

1) The most serious concern is on the novelty of the current study. In the main text, the authors wrote "However, previous attempts at electric-field-driven metallization of VO₂ have not been successful", but this is not true. At least I am aware of some papers from the Japanese group [T. Yajima, et. al., Nat. Commun. 6, 10104 (2015); T. Yajima, et. al., Adv. Electron. Mater. 6, 1901356 (2020); T. Yajima, et. al., Adv. Electron. Mater. 8, 2100842 (2022)], where the authors performed very systematic and thorough experiments on the electric-field effects on the MIT properties of VO₂ by using a solid gate dielectric (either TiO₂ or HfO₂ was used as a gate dielectric layer) and demonstrated the modulation of T_{MIT} by electron doping without introducing structural distortion. Considering that the main significance of the current study seems to be the same as what those previous studies have already demonstrated (the modulation of the MIT by electron doping without introducing structural distortion), I do not find a sufficient novelty in the current study that would satisfy the criteria for publication in Nature Communications. Moreover, similar "modulation doping" experiments were also performed by the same group [T. Yajima, et. al., Small 13, 1603113 (2017)], where the authors prepared the heterostructures with pristine VO₂ and electron-doped VO₂, and examined the impact of the "modulation doping" effect on the MIT properties in a very sophisticated manner. Those previous studies might decrease the overall value of the present study.

2) The second concern is on the interpretation of their observations. The authors argue that there was no change in the structural properties, but it is not well supported by the experimental data. They wrote "Based on the q -2 q XRD measurements, reciprocal space maps, and cross-sectional STEM imaging we conclude that the lattice parameter changes, if any, are within the instrumental resolution (better than 0.1 pm) for all the VO₂ heterostructures used in this work", but the characterizations that the authors did in the manuscript are not enough. For example, from the q -2 q XRD data, they should be able to deduce the numerical values of the out-of-plane lattice parameters, but they did not provide those values in the manuscript. The discussion on the presence

or absence of the lattice distortion should be done in a more scientific way based on the experimental data. At least it looks to me that there is some variation in the out-of-plane lattice parameter depending on the thickness, suggesting that the lattice distortion should play a role in their observations. The same is true for the reciprocal space mapping data. For the cross-sectional STEM imaging, I could only find one STEM image in Fig. 1c, and I could not find any information that supports their claim written above.

3) The third one is on the origin of the shift of T_{MIT} . The authors insist that the electron doping should purely originate from "modulation doping" from TiO_{2-x} , but I think it is not so conclusive. As the authors wrote in the manuscript, there is a possibility of the oxygen-vacancy formation in VO_2 during the sample preparation, which in general reduces T_{MIT} quite a lot. Based on the XPS results, the authors mentioned "However, the spectra do not show any signatures of oxygen vacancies in VO_2 heterostructures. Therefore, we rule out any oxygen-vacancy induced carrier doping in these heterostructures", but it is very weak as experimental evidence. To be fair, it looks at least to me that many experimental data support the idea that the oxygen-vacancy formation should play a major role for their observations. For example, if the modulation doping plays a predominant role, the doping effect should be limited to the interface between VO_2 and $LaAlO_3$. In such a case, the MIT should have two components associated with the bulk region having higher T_{MIT} and the interface region hosting lower T_{MIT} . However, in reality, the authors observed the single component MIT for all the samples. This strongly suggests that the oxygen-vacancy formation should play a major role. Moreover, the fact that the shift of T_{MIT} was observed even for the heterostructures without TiO_{2-x} also supports this oxygen-vacancy formation scenario. To rule out this possibility, the authors should perform another control experiment, the spacer-layer-thickness dependence of the MIT behavior. If the modulation doping is at work, the shift of T_{MIT} should be decreased systematically with increasing the thickness of the spacer layer. I think this is a necessary experiment to support their claim.

Reviewer #3 (Remarks to the Author):

Review comments for NCOMMS-23-04444-T

This paper demonstrates the modulation-doped VO_2 -based thin film heterostructures to prove the filling controlled-metal-insulator transition (MIT) phenomenon. Considering the filling controlled-MIT is a main operating mechanism in a Mott-tronics for accelerating the MIT behavior, the scientific proof based on material analyses is sufficiently valuable to be published in this Nature Communications journal. However, there are some critical points needed to be revised. If the authors of this paper conduct some major revisions, it will be published in this journal.

1. In the Introduction part, the authors mention the external stimuli that induce the MIT behavior in a VO₂. In this sentence (“In particular, there is widespread interest in the nature of the insulating state and its external control via doping, strain, oxygen vacancy creation, hydrogenation, light-and-pulse-induced modulation, and via electric fields in a field-effect transistor geometry”), please add a reference in the part of “electric fields in a field-effect transistor geometry”. Recently, a new paper about a hetero-epitaxial Mott transistor based on BiFeO₃ ferroelectrics was published. Please refer this paper “ <https://onlinelibrary.wiley.com/doi/10.1002/adma.202203097>”.

2. The authors in this paper use the spacer layer as LAO. As mentioned in page 4, the bandgap of LAO is high enough to circumvent the oxygen vacancy diffusion of VO₂. However, the thickness of LAO is 2 nanometers, which is absolutely small enough to induce a current tunneling. Furthermore, since its device structure is not based on an epitaxial heterostructure, which alludes vulnerability of interface defects, the trap sites generated at the VO₂/LAO may lead to the tunneling routes. I think this kind of carrier path can increase the carrier mobility when the Hall measurements of heterostructures were measured. As shown in Figure 3b, the carrier mobility of 9.5 VO₂ – het is larger than of 9.5 VO₂ near the room temperature. This result may be also related with the LAO tunneling effect, even if it's trivial. Please provide some rebuttal explanations or suitable references about this assumption.

3. Authors of this paper alleged that the structural phase transition didn't occur across the filling controlled-MIT phenomenon. However, some kinds of structural analyses during the Hall effect measurement cannot be found in this paper. Could you provide a structural analysis when the MIT occurs? It's okay if the analysis is conducted using the heterostructures.

4. The thickness-varied VO₂ samples might definitely be affected by a lattice strain. Although all the thicknesses of VO₂ are below 10 nanometers, the lattice strain must have been more applied as the thickness of it is thinner. Please provide some supporting explanation that the lattice strain affects the carrier density.

5. As shown in Figure 4b, the different main peaks in the 7.5 VO₂ and 7.5 VO₂-het samples, which are 514.5 eV and 517.5 eV respectively, imply that the band bending of VO₂-het helps to induce the higher oxidation states. This kind of oxidation state shift cannot be explained with this enhanced carrier density trend. The appearance of V⁵⁺ peak (517.5 eV) should be explained with reasons of the improved carrier density in the heterostructure geometry. If this explanation cannot be provided, authors must provide this appearance of V⁵⁺ binding peak in XPS data.

Manuscript Number: NCOMMS-23-04444-T

Title: Modulation-Doping a Correlated Electron Insulator

Authors: Debasish Mondal¹, Smruti Rekha Mahapatra¹, Abigail M Derrico², Rajeev Kumar Rai³, Jay R Paudel², Christoph Schlueter⁴, Andrei Gloskovskii⁴, Rajdeep Banerjee¹, Atsushi Hariki⁵, Frank M F DeGroot⁶, Dipankar D Sarma¹, Awadhesh Narayan¹, Pavan Nukala³, Alexander X Gray^{2*} and Naga Phani B Aetukuri^{1*}

RESPONSE TO REVIEWER COMMENTS

We thank all the reviewers for their time and interest in our work. We are pleased to note that all reviewers, in general, recommended the publication of our manuscript after addressing the suggested changes. Below, we address the reviewers' comments. Reviewers' comments are in blue-colored font and our responses are in black-colored font. Changes made to the text in the main manuscript or the supplementary text are highlighted in yellow.

Reviewer #1 (Remarks to the Author):

This paper presents a systematic study combining the film synthesis, novel doping strategy and characterization (structure, composition and electrical) of VO₂ films. Given that VO₂ is one of the condensed matter systems that is heavily studied, it is perhaps worth noting that authors managed to discover a novel strategy to study electronic vs structural transition. It is further noteworthy (at least in this reviewer's opinion) that the use of amorphous layer of ultra wide bandage oxide LaAlO₃ as a space layer and amorphous oxygen deficient TiO₂ as a donor layer yields a controlled modulation doping strategy. As authors have pointed out this strategy eliminate the issue pertaining to lattice mismatch, and oxygen vacancy formation in VO₂ as well as eliminate the concerns related to transport owing to intermixing. Note that both La and Al on LaAlO₃ are not electron donors in VO₂. Using a range of approaches combining transport, hard x-ray photoelectron spectroscopy, and structural characterization, authors demonstrated modulation doping of VO₂. High density of electron was achieved which allowed authors to control electronic transition (insulator to metal) without disturbing the structure. Results presented in Fig 2c is particularly noteworthy where a large change in T_{MIT} is achieved with no change in the lattice parameters. The methodology using amorphous layer

for modulation doping should also be relevant to other materials systems and hence, will likely be of broader interest.

Paper is very well written, and data is nicely presented. I recommend publication of the manuscript as is.

We thank the reviewer for their positive assessment of our work and its broader implications for modulation-doping correlated electron insulators.

Reviewer #2 (Remarks to the Author):

The authors report on the effect of "modulation doping" on the metal-insulator transition (MIT) properties of VO₂ in the VO₂-based heterostructures. VO₂ is one of representative strongly-correlated oxides, showing the electronic phase transition (i.e., MIT) near room temperature. This MIT is known to accompany the structural phase transition, and there has been a long-standing debate on the driving force of this phase transition, in particular on whether the electronic transition and the structural transition could be decoupled or not. To address this issue, the authors prepared the VO₂-based heterostructures with an insulating spacer layer made of LaAlO₃ and a carrier reservoir layer made of oxygen deficient TiO_{2-x}, and examined the impact of the charge-transfer doping from TiO_{2-x} to VO₂ on the MIT properties of VO₂. Then, based on the experimental results including XRD, STEM, transport, and HAXPES measurements, the authors conclude that they could modulate the MIT temperature (T_{MIT}) purely by electron doping without introducing structural deformation. They observed finite reduction of T_{MIT} together with the suppression of the resistance at the low-temperature insulating phase, both of which appeared to be associated with the electron doping as they in fact confirmed by the Hall-effect measurements. They also found that this effect became stronger as the thickness of the VO₂ layer was reduced. I agree that the electron doping in VO₂ should take place in their heterostructures, but I have a concern on their interpretation. I also have other concerns on the present study, which should preclude publication of this study, at least with the current version. The authors should address the following concerns in full before further consideration.

We thank the reviewer for their interest in our work and for their review of our work. Below, we give a point-by-point response to their comments.

1) The most serious concern is on the novelty of the current study. In the main text, the authors

wrote "However, previous attempts at electric-field-driven metallization of VO₂ have not been successful", but this is not true. At least I am aware of some papers from the Japanese group [T. Yajima, et. al., Nat. Commun. 6, 10104 (2015); T. Yajima, et. al., Adv. Electron. Mater. 6, 1901356 (2020); T. Yajima, et. al., Adv. Electron. Mater. 8, 2100842 (2022)], where the authors performed very systematic and thorough experiments on the electric-field effects on the MIT properties of VO₂ by using a solid gate dielectric (either TiO₂ or HfO₂ was used as a gate dielectric layer) and demonstrated the modulation of T_{MIT} by electron doping without introducing structural distortion. Considering that the main significance of the current study seems to be the same as what those previous studies have already demonstrated (the modulation of the MIT by electron doping without introducing structural distortion), I do not find a sufficient novelty in the current study that would satisfy the criteria for publication in Nature Communications. Moreover, similar "modulation doping" experiments were also performed by the same group [T. Yajima, et. al., Small 13, 1603113 (2017)], where the authors prepared the heterostructures with pristine VO₂ and electron-doped VO₂, and examined the impact of the "modulation doping" effect on the MIT properties in a very sophisticated manner. Those previous studies might decrease the overall value of the present study.

By the statement "However, previous attempts at electric-field-driven metallization of VO₂ have not been successful", we wanted to convey that previous research efforts to metalize VO₂ using external electric fields did not modulate the transition temperature (T_{MIT}) by a significant amount. The papers shared by the reviewer further reinforce this argument. For example, the papers shared by the reviewer show at best ~1 K reduction in T_{MIT} at gate voltages as high as ~9 V.¹⁻³ Whereas, in this study, we show that modulation-doping enables a > 60 K reduction in T_{MIT}. This has no precedence in published literature.

Also, the contents of the work in a fourth reference⁴ shared by the referee are fundamentally different from our work. In the work cited by the reviewer, epitaxial heterostructures comprise of tungsten-doped VO₂ (W_xV_{1-x}O₂), wherein the transition temperature is modulated by a change in the elemental dopant's (tungsten in this case) concentration.

By stark contrast, our work concerns electron-doping by exploiting electronic chemical potential mismatch at heterointerfaces. Furthermore, we deliberately used a 2 nm LaAlO₃ spacer layer as a diffusion barrier to kinetically limit any elemental interdiffusion from the dopant layer.

Therefore, the results of our work have no precedence in published literature. Furthermore, we believe that this technique could be applied to other correlated insulators and enable the experimental realization of filling-controlled metal-insulator transitions.

To further clarify this in the manuscript we updated the main manuscript with the following additional references:

1. Yajima, T., Nishimura, T. & Toriumi, A. Positive-bias gate-controlled metal–insulator transition in ultrathin VO₂ channels with TiO₂ gate dielectrics. *Nat Commun* 6, 10104 (2015).
2. Yajima, T. & Toriumi, A. Observation of the Pinch-Off Effect during Electrostatically Gating the Metal-Insulator Transition. *Advanced Electronic Materials* 8, 2100842 (2022).
3. Yajima, T., Nishimura, T. & Toriumi, A. Identifying the Collective Length in VO₂ Metal–Insulator Transitions. *Small* 13, 1603113 (2017).

We have also modified the text in paragraph 5 of page 2 to paragraph 1 of page 3 as follows: We removed the sentence: “*However, previous attempts at electric-field-driven metallization of VO₂ have not been successful.*” and modified the paragraph as:

“However, previous attempts at electric-field-driven metallization of VO₂ were not successful. For example, attempts at modulating the MIT in VO₂ in a field effect transistor geometry yielded less than a 1 K change in T_{MIT}. The weak response of T_{MIT} of VO₂ to external electric field, even when gated through high-K dielectrics, was attributed to the presence of strong correlations in the insulating VO₂ phase. Further, ionic-liquid gating of VO₂, which could enable accessibility to large interfacial electric-fields, led to oxygen vacancy creation and/or hydrogenation of VO₂. Heterostructures with differing compositions such as for example VO₂/W_xV_{1-x}O₂ based heterostructure thin films showed a larger change in T_{MIT}. However, these changes are related to elemental doping driven by a chemical potential mismatch of the dopant-ion (W⁶⁺ in this specific case). By stark contrast, we propose modulation-doping of VO₂ using electronic chemical potential differences at oxide heterostructures.”

2) The second concern is on the interpretation of their observations. The authors argue that there was no change in the structural properties, but it is not well supported by the experimental data. They wrote "Based on the q-2q XRD measurements, reciprocal space maps, and cross-sectional STEM imaging we conclude that the lattice parameter changes, if any, are within the instrumental resolution (better than 0.1 pm) for all the VO₂ heterostructures used in this work",

but the characterizations that the authors did in the manuscript are not enough. For example, from the q - $2q$ XRD data, they should be able to deduce the numerical values of the out-of-plane lattice parameters, but they did not provide those values in the manuscript. The discussion on the presence or absence of the lattice distortion should be done in a more scientific way based on the experimental data. At least it looks to me that there is some variation in the out-of-plane lattice parameter depending on the thickness, suggesting that the lattice distortion should play a role in their observations. The same is true for the reciprocal space mapping data. For the cross-sectional STEM imaging, I could only find one STEM image in Fig. 1c, and I could not find any information that supports their claim written above.

We thank the reviewer for this question. We agree that the statement about the resolution of the measurements could be confusing. To address this question, we have now tabulated out-of-plane lattice parameters of all the thin films and heterostructures along with the measured transition temperatures. We have also added columns to clearly show how the parameters ΔC_R and ΔT_{MIT} are calculated.

While there is some apparent change in the position of XRD reflection as a function of VO_2 thickness, this is not consistent with strain relaxation. As the film thickness decreases, there is finite thickness broadening which increases the width and decreases the intensity of the VO_2 (002) reflection. Consequently, there is a greater contribution to this reflection from the tail of the TiO_2 (002) substrate reflection. This leads to an apparent shift of the VO_2 (002) reflection towards the substrate's reflection (towards lower 2θ values).

To avoid any misinterpretation of data related to this apparent shift in the XRD reflections induced by the substrate background, we measured θ - 2θ plots within the same angular range for both thin films and heterostructures with an identical thickness of VO_2 . These plots are presented in Supplementary Fig. 4. These plots clearly show that there is no difference in the characteristics of the XRD reflections due to heterostructuring.

Since VO_2 is tensile-strained in the plane of the film ($a_{VO_2} = 4.55 \text{ \AA}$; $a_{TiO_2} = 4.59 \text{ \AA}$), a decrease in the out-of-plane lattice parameter is expected for coherently strained VO_2 films. This implies that the thinnest films, with little to no strain relaxation, must have the smallest out-of-plane lattice parameter. Any strain relaxation, as the thickness increases, must increase the out-of-plane lattice parameter leading to an *angular down shift in the XRD peak for thicker films*. The apparent angular down shift for heterostructures with thinner VO_2 in the XRD data shown in Figure 2a of the original manuscript is inconsistent with strain relaxation.

We would also like to note that our conclusions are consistent with previously published literature. It is well known that VO₂ films are coherently strained to the TiO₂ (001) substrate up to a critical thickness of ~16 nm.^{5,6} Since all the films shown in this study are below 16 nm, we expect a constant strain for all the VO₂ films. Consistent with this, reciprocal space maps on all the heterostructures prepared in this work show that VO₂ is coherently strained to the underlying TiO₂ (001) substrate. This is also consistent with cross-section STEM data performed for the VO₂ film and a VO₂ heterostructure. We have now added this comparison from STEM imaging in the supplementary material (see Supplementary Fig. 1 in the revised version) [also please see asymmetrical RSM images in Supplementary Fig. 5 in the revised supplementary section]. Therefore, we conclude that any changes to the XRD peak position cannot be due to strain relaxation but might be related to background contribution from the substrate background as the film thickness decreases.

However, at lower film thicknesses, the total intensity of the VO₂ film reflection was too weak to accurately disentangle the intensity contributions from the TiO₂ substrate reflection and the VO₂ reflection and estimate lattice parameters. Therefore, we deposited films and heterostructures of the same thickness and compared the changes in the out-of-plane lattice parameter (C_R) and the T_{MIT} for the film and heterostructure with the same VO₂ thickness. This data is now presented in Supplementary Table 1 and used to plot Fig. 2c. It can be seen that the out-of-plane lattice parameters for both the film and heterostructures are nearly identical for the same thickness of VO₂. Hence the differential out-of-plane lattice parameters (ΔC_R) are independent of the VO₂ thickness suggesting there are no changes in strain due to the formation of heterostructures. This is of importance for this work, as the change in T_{MIT} is brought about by heterostructure formation. The errors in the calculation of the lattice parameter if any are due to the broadening of the VO₂ film's reflection as the thickness is decreased.

For the referee's and readers' convenience, we have now added a table to the supplementary section [Supplementary Table 1], with lattice parameters extracted from the XRD data by curve fitting. The data from this table was used in Fig. 2c.

To further clarify this in the manuscript we have also added the STEM image in Supplementary Fig. 1 in the revised version of the supplementary. we reproduced these figures here for the referee's convenience (see Figure R1 and Table R1) and added the associated sentence in paragraph 1 of page 5 in the main text:

“Furthermore, the STEM image of pristine VO₂ on TiO₂ (001) substrate shows (supplementary Fig. 1) that the epitaxial atomic arrangement of VO₂ is identical for VO₂ heterostructures and thin films.”

And modified the sentence in paragraph 3 of page 5 to paragraph 1 of page 6 in the main text to:

“Based on θ -2 θ XRD measurements, reciprocal space maps, and cross-sectional STEM imaging we conclude that there are no changes in the lattice parameters between VO₂ films and heterostructures. And therefore, changes in strain cannot account for the reduction in T_{MIT} observed in modulation doped VO₂ heterostructures (see Supplementary Figures. 1 and 4 and Supplementary Table 1).”

We also added Supplementary Fig. 1 and Supplementary Table 1, which are reproduced here for the reviewer’s convenience.

Figure R1. Room temperature STEM images of (a) 6 nm VO₂ on TiO₂ (001) substrate and (b) 6 nm VO₂/ 2 nm LAO/ 3 nm TiO_{2-x}/1 nm LAO deposited on TiO₂ (001) substrate. In both cases, VO₂ is coherently strained to the TiO₂ (001) substrate.

VO ₂ thickness (nm)	Extracted out-of-plane lattice parameter (C _R) from XRD fitting (pm)		$\Delta C_R = (C_{R-het} - C_{R-film})$ (pm)	Transition temperature (T _{MIT}) (K)		$\Delta T_{MIT} = (T_{MIT-het} - T_{MIT-film})$ (K)
	VO ₂ film (C _{R-film})	VO ₂ heterostructure (C _{R-het})		VO ₂ film (T _{MIT-film})	VO ₂ heterostructure (T _{MIT-het})	
9.5	283.33	283.34	0.01	295	282	-13
7.5	283.24	283.25	0.01	295	275	-20
4.5	283.23	283.24	0.01	295	260	-35
3.5	284.11	284.11	0	295	250	-45
2.5	284.28	284.28	0	297	237	-60
1.5	284.29	284.29	0	299	233	-66

Table. R1. A comparison of out-of-plane lattice parameters (c_R) and T_{MIT} between VO_2 film and VO_2 heterostructures. LEPTOS 7.8 from Bruker was used to extract the lattice parameters using a pseudo-voigt function for all the films and heterostructures. Based on q - $2q$ X-ray diffractograms in Fig. 2a, there is an apparent downshift in the position of the VO_2 (002) with decreasing film thickness. We note that this downshift in the angular position of the VO_2 (002) reflection with decreasing film thickness is inconsistent with strain relaxation which should downshift the VO_2 (002) reflection's angular position with increasing thickness. This is because the in-plane lattice parameters of TiO_2 (001) ($a_{TiO_2} = 4.59 \text{ \AA}$) are greater than bulk VO_2 (001) ($a_{VO_2} = 4.55 \text{ \AA}$). Strain relaxation, if any, (possible with increasing thickness) should decrease out-of-plane compression and increase the out-of-plane lattice parameter (leading to a downshift in the angular position of VO_2 (002) reflection) with increasing VO_2 thickness. Therefore, we hypothesized that the apparent downshift may be due to an increased contribution from the TiO_2 (002) substrate reflection to the VO_2 (002) reflection in heterostructures with thinner VO_2 . However, at lower film thicknesses, the total intensity of the VO_2 film reflection was too weak to accurately disentangle the intensity contributions from the TiO_2 substrate reflection and the VO_2 reflection. Therefore, we deposited films and heterostructures of the same thickness and compared the changes in the out-of-plane lattice parameters and the T_{MIT} across the film and heterostructure with the same VO_2 thickness. It can be seen that the out-of-plane lattice parameters for both the film and heterostructures are nearly identical for the same thickness of VO_2 . Hence the differential out-of-plane lattice parameters (ΔC_R) are independent of the VO_2 thickness suggesting there are no changes in strain due to the formation of heterostructures. The errors in the calculation of the lattice parameter if any are due to the broadening of the VO_2 film's reflection as the thickness is decreased. Our conclusions are also consistent with previous literature reports^{5,6} that show VO_2 (001) films are coherently strained to TiO_2 (001) substrates up to a critical thickness of 16 nm. We note that all VO_2 films in this work are therefore restricted to thicknesses of <10 nm to avoid any strain-related shifts in T_{MIT} .

3) The third one is on the origin of the shift of T_{MIT} . The authors insist that the electron doping should purely originate from "modulation doping" from TiO_{2-x} , but I think it is not so conclusive. As the authors wrote in the manuscript, there is a possibility of the oxygen-vacancy formation in VO_2 during the sample preparation, which in general reduces T_{MIT} quite a lot. Based on the XPS results, the authors mentioned "However, the spectra do not show any

signatures of oxygen vacancies in VO₂ heterostructures. Therefore, we rule out any oxygen-vacancy induced carrier doping in these heterostructures", but it is very weak as experimental evidence. To be fair, it looks at least to me that many experimental data support the idea that the oxygen-vacancy formation should play a major role for their observations. For example, if the modulation doping plays a predominant role, the doping effect should be limited to the interface between VO₂ and LaAlO₃. In such a case, the MIT should have two components associated with the bulk region having higher T_{MIT} and the interface region hosting lower T_{MIT}. However, in reality, the authors observed the single component MIT for all the samples. This strongly suggests that the oxygen-vacancy formation should play a major role. Moreover, the fact that the shift of T_{MIT} was observed even for the heterostructures without TiO_{2-x} also supports this oxygen-vacancy formation scenario. To rule out this possibility, the authors should perform another control experiment, the spacer-layer-thickness dependence of the MIT behavior. If the modulation doping is at work, the shift of T_{MIT} should be decreased systematically with increasing the thickness of the spacer layer. I think this is a necessary experiment to support their claim.

We thank the reviewer for this excellent suggestion. We performed these additional control experiments. We performed experiments with a spacer layer thickness of 2 nm, 4 nm, and 10 nm, and the results are summarized in Figure R2. The figure shows that increasing the thickness of the spacer layer increases T_{MIT} suggestive of a decreased charge transfer from the TiO_{2-x} dopant layer to the VO₂ channel layer – exactly as expected for tunneling of carriers across the spacer layer in modulation-doping. These additional experiments further reinforce our original proposition that the shift in the transition temperature is enabled by modulation-doping.

In addition, we would like to note that both XPS and XRD were shown to be extremely sensitive to the presence of oxygen vacancies.^{7,8} For example, a huge increase (by more than a few percent, about >3 pm) in lattice parameters of VO₂ films was reported by several groups, for oxygen-deficient VO₂.^{8,9} We did not observe any such change in the lattice parameters across films and heterostructures of the same thickness in XRD measurements.

Furthermore, a distinct V³⁺ signature was reported in XPS measurements performed on oxygen-deficient VO₂,⁷ suggesting that XPS is sensitive to detect oxygen vacancies. If at all oxygen vacancies existed in the VO₂ heterostructures deposited in this work, we must have seen V³⁺, especially at the carrier densities that we realized. However, we did not observe any signatures of V³⁺ in XPS measurements performed on VO₂ heterostructures.

Clearly, we did *not* observe the presence of V^{3+} or any increase in the lattice parameter of the VO_2 phase. Put together, XPS and XRD measurements are unequivocal evidence that elemental doping or oxygen vacancies could not have induced the observed decrease in T_{MIT} .

A second point that the reviewer raised is the absence of a double transition – one for the interfacial region and the other for the rest of the film. In general, carrier delocalization length scales determine whether a second transition will be observed or not, and therefore the observation of a double transition is *not* a necessary condition for modulation-doping. We note that all films used in this study are < 10 nm and therefore it is plausible that the carriers are delocalized over this entire length. Both hall measurements and transport are consistent with carrier delocalization. In fact, this is an intriguing feature of this study. We note that the possibility of bulk destabilization of a Mott state with interfacial doping has been proposed before.¹⁰ We would also like to note that the references^{1,2} pointed out by this referee also show this phenomenon, where there appears to be a single transition in electric field-gated devices.

While the absence of two transitions is intriguing and needs further study, this points to an exciting aspect of Mott Physics. And provides further support for the utility of our work in the broader field of electron doping in correlated electron insulators.

Figure R2: MIT characteristics of VO_2 Modulation doped heterostructures as a function of the LAO spacer-layer-thickness thicknesses mentions in the figure legend. In the figure legend, as an example, $7.5VO_2$ -het (2 nm LAO) represents a VO_2 heterostructure with 7.5 nm VO_2 / 2 nm LAO/ 3 nm TiO_{2-x} / 1 nm LAO.

Reviewer #3 (Remarks to the Author):

Review comments for NCOMMS-23-04444-T

This paper demonstrates the modulation-doped VO₂-based thin film heterostructures to prove the filling controlled-metal-insulator transition (MIT) phenomenon. Considering the filling controlled-MIT is a main operating mechanism in a Mott-tronics for accelerating the MIT behavior, the scientific proof based on material analyses is sufficiently valuable to be published in this Nature Communications journal. However, there are some critical points needed to be revised. If the authors of this paper conduct some major revisions, it will be published in this journal.

We thank the reviewer for their positive assessment of our work and for their interest in our work.

1. In the Introduction part, the authors mention the external stimuli that induce the MIT behavior in a VO₂. In this sentence (“In particular, there is widespread interest in the nature of the insulating state and its external control via doping, strain, oxygen vacancy creation, hydrogenation, light-and-pulse-induced modulation, and via electric fields in a field-effect transistor geometry”), please add a reference in the part of “electric fields in a field-effect transistor geometry”. Recently, a new paper about a hetero-epitaxial Mott transistor based on BiFeO₃ ferroelectrics was published. Please refer this paper “<https://onlinelibrary.wiley.com/doi/10.1002/adma.202203097>”.

We thank the referee for sharing the relevant reference. We cited the reference for “electric fields in a field-effect transistor geometry” in the main text.

2. The authors in this paper use the spacer layer as LAO. As mentioned in page 4, the bandgap of LAO is high enough to circumvent the oxygen vacancy diffusion of VO₂. However, the thickness of LAO is 2 nanometers, which is absolutely small enough to induce a current tunneling. Furthermore, since its device structure is not based on an epitaxial heterostructure, which alludes vulnerability of interface defects, the trap sites generated at the VO₂/LAO may lead to the tunneling routes. I think this kind of carrier path can increase the carrier mobility when the Hall measurements of heterostructures were measured. As shown in Figure 3b, the carrier mobility of 9.5 VO₂ – het is larger than of 9.5 VO₂ near the room temperature. This

result may be also related with the LAO tunneling effect, even if it's trivial. Please provide some rebuttal explanations or suitable references about this assumption.

The referee is correct that the tunneling of carriers from the n-doped TiO_{2-x} layer across the LAO barrier is the expected mechanism for modulation-doping VO_2 . Please also refer to the response to Question 3 of Reviewer 2, where we show that charge transfer across the LAO barrier decreases as the thickness of the barrier increases.

However, tunneling may not be the origin of the increase in mobility that the reviewer points to. In general, in the insulating state, mobility decreases with increased carrier concentration for all the VO_2 samples used in this work. In the metallic state, the carrier densities and the associated carrier mobilities of all the samples are nearly identical. This is along expected lines. The specific data point, mobility of 9.5 nm thin film and thin film heterostructure at 290 K, that the referee identified is a special case. At 290 K, 9.5 nm VO_2 is in the insulating state, while 9.5 nm VO_2 -het is in the metallic state. Therefore, the mobilities are not directly comparable. Any comparison of these two samples at the temperature of 290 K is not correct. We note that the mobilities are nearly identical when both the samples are in the metallic state (>300 K). Furthermore, in the insulating state, the 9.5 nm thin film with a lower carrier density has a higher carrier mobility than the 9.5 nm VO_2 -het which has a higher carrier density (and relatively lower carrier mobility).

3. Authors of this paper alleged that the structural phase transition didn't occur across the filling controlled-MIT phenomenon. However, some kinds of structural analyses during the Hall effect measurement cannot be found in this paper. Could you provide a structural analysis when the MIT occurs? It's okay if the analysis is conducted using the heterostructures.

It seems that there is some confusion. We have never claimed that we suppressed the structural phase transition. We wanted to convey the message that in the case of filling controlled MIT of VO_2 , the phase transition temperature (T_{MIT}) should purely be controlled by the electron densities without any associated changes to the lattice parameters of the parent phases. The phase transition (the simultaneous structural and electronic phase transition) would still occur, albeit at a lowered temperature as the electronic density in the insulating state increases. Supplementary Fig. 4 in SI shows that the lattice parameter of VO_2 (rutile phase) is identical for the heterostructure and the thin film.

To further clarify the reviewer’s question and check for the presence of a structural phase transition in heterostructures, we performed temperature-dependent XRD measurements at 320 K (metallic phase) and 200 K (insulating phase) for a 9.5 nm VO₂ thin film and 9.5 nm and 4.5 nm VO₂ heterostructures. Figure R3a shows the signature of the structural phase transition from the rutile phase (at 320 K) to the monoclinic phase (at 200 K) for the 9.5 nm VO₂ film. Please note that in Figure R3, the reflection referred to as ($\bar{4}02$)_M is the Bragg reflection peak in the monoclinic phase of VO₂. In the low-temperature monoclinic phase the ($\bar{4}02$)_M peak appears at a lower 2θ angle than the Bragg angle of (002)_R peak in the rutile phase. Similar spectral features were also observed for the 9.5VO₂-het and 4.5VO₂-het as well suggesting that the MIT is being accompanied by the structural phase transition (see Figures R3b and c).

Figure R3. Temperature-dependent XRD spectra of (a) 9.5VO₂, (b) 9.5VO₂-het, and (c) 4.5VO₂-het were measured at both the insulating phase (at 200 K) and the metallic phase (at 320 K). A clear peak shift of VO₂ was observed across the MIT. The ($\bar{4}02$)_M peak is the Bragg reflection peak in the monoclinic phase of VO₂ which is equivalent to (002)_R peak in the rutile phase.

Additionally, in Figure R4 we have also compared the XRD spectra of 9.5VO_2 and $9.5\text{VO}_2\text{-het}$ both in the insulating phase (at 200 K) and metallic phase (at 320 K). It clearly shows there is an excellent overlap of the two diffractograms, including thickness fringes, for both structural phases providing further evidence that there are no measurable changes to the structure. This further rules out oxygen vacancies or other elemental dopants (and their associated strain effects) as possible sources of charge doping.

Figure R4. An overlap of XRD spectra between 9.5VO_2 and $9.5\text{VO}_2\text{-het}$ at (a) 200 K and (b) 320 K. The $(\bar{4}02)_M$ peak is the Bragg reflection peak in the monoclinic phase of VO_2 which is equivalent to the $(002)_R$ peak in the rutile phase.

4. The thickness varied VO_2 samples might definitely be affected by a lattice strain. Although all the thicknesses of VO_2 are below 10 nanometers, the lattice strain must have been more applied as the thickness of it is thinner. Please provide some supporting explanation that the lattice strain affects the carrier density.

We performed careful XRD analysis to rule out any thickness-dependent strain-induced changes to the transition temperature. While there is some apparent change in the position of XRD reflection as a function of VO_2 thickness, this is not consistent with strain relaxation [please see Figure 2c and Supplementary Figure 1 and Table 1 in the revised supplementary file]. As the film thickness decreases, there is finite thickness broadening which increases the width and decreases the intensity of the VO_2 (002) reflection. Consequently, there is a greater contribution to this reflection from the tail of the TiO_2 (002) substrate reflection. This leads to

an apparent shift of the VO₂ (002) reflection towards the substrate's reflection (towards lower 2θ values).

To avoid any misinterpretation of data related to this apparent shift in the XRD reflections induced by the substrate background, we measured θ-2θ plots within the same angular range for both thin films and heterostructures with an identical thickness of VO₂. These plots, in Supplementary Fig. 4, clearly show that there is no difference in the characteristics of the XRD reflections due to heterostructuring. The changes in the lattice parameters, if any, are within the instrumental resolution of our experiments. Based on the data presented in Fig. 2c in the main text and Supplementary Table 1 and Supplementary Fig. 1 in the revised version of the supplementary, the decrease in T_{MIT} in heterostructures cannot be due to strain or elemental doping.

We would also like to request the referee to refer to a more detailed response provided to a similar question by Reviewer 2 [Question 2 from Reviewer 2]

5. As shown in Figure 4b, the different main peaks in the 7.5 VO₂ and 7.5 VO₂-het samples, which are 514.5 eV and 517.5 eV respectively, imply that the band bending of VO₂-het helps to induce the higher oxidation states. This kind of oxidation state shift cannot be explained with this enhanced carrier density trend. The appearance of V⁵⁺ peak (517.5 eV) should be explained with reasons of the improved carrier density in the heterostructure geometry. If this explanation cannot be provided, authors must provide this appearance of V⁵⁺ binding peak in XPS data.

Our results suggest that the origin of this peak may not be due to V⁵⁺. First, as we mentioned in the originally submitted main manuscript, an increase in the oxidation state from V⁴⁺ to V⁵⁺ cannot explain the observed electron doping in VO₂ heterostructures since electron doping should decrease the V⁴⁺ oxidation state in VO₂. Additionally, we also addressed in the main manuscript that there is a strong correlation between this peak (at 517.5 eV) intensity and the carrier doping in VO₂ (please see Fig. 3a and Figs. 4a and b in the main manuscript).

Furthermore, in this revised version we (A. Hariki) performed detailed LDA+DMFT-Anderson Impurity calculations of VO₂ as a function of doping to determine the origin of the 517.5 eV peak. The impurity calculation of the 2p XPS spectrum of VO₂ shows that electron doping creates a satellite peak at approximately 517.5 eV. Based on this calculation we assign the 517.5 eV peak to a satellite induced by electron doping. We added the calculations in the revised version of the supplementary information (see Supplementary Fig. 16).

Also, we added the following text in the main manuscript on page 10 in paragraph 3:

“Furthermore, since there is a strong correlation between electron density and $P2$ peak intensity, we performed an LDA+DMFT-Impurity calculation for the undoped and electron-doped VO_2 to examine the $V2p$ XPS spectral changes with electron doping. The calculation of the $V2p$ XPS spectrum shows that electron doping creates a satellite peak at approximately 517.5 eV (see Supplementary Note 2 and Supplementary Fig. 16 for details of the calculations). Since the V-O covalency for the V^{3+} (d^2) state is weaker than V^{4+} (d^1) one, the binding energy of V^{3+} in electron-doped VO_2 increases due to a weaker bonding and anti-bonding splitting in the XPS final states. Therefore, we assign the $P2$ peak to a satellite induced by electron doping.”

References:

1. Yajima, T., Nishimura, T. & Toriumi, A. Positive-bias gate-controlled metal–insulator transition in ultrathin VO_2 channels with TiO_2 gate dielectrics. *Nat Commun* **6**, 10104 (2015).
2. Yajima, T. & Toriumi, A. Observation of the Pinch-Off Effect during Electrostatically Gating the Metal-Insulator Transition. *Advanced Electronic Materials* **8**, 2100842 (2022).
3. Yajima, T., Nishimura, T., Tanaka, T., Uchida, K. & Toriumi, A. Modulation of VO_2 Metal–Insulator Transition by Ferroelectric HfO_2 Gate Insulator. *Advanced Electronic Materials* **6**, 1901356 (2020).
4. Yajima, T., Nishimura, T. & Toriumi, A. Identifying the Collective Length in VO_2 Metal–Insulator Transitions. *Small* **13**, 1603113 (2017).
5. Nagashima, K., Yanagida, T., Tanaka, H. & Kawai, T. Stress relaxation effect on transport properties of strained vanadium dioxide epitaxial thin films. *Phys. Rev. B* **74**, 172106 (2006).
6. Fan, L. L. *et al.* Strain Dynamics of Ultrathin VO_2 Film Grown on TiO_2 (001) and the Associated Phase Transition Modulation. *Nano Lett.* **14**, 4036–4043 (2014).

7. Jeong, J. *et al.* Suppression of Metal-Insulator Transition in VO₂ by Electric Field-Induced Oxygen Vacancy Formation. *Science* **339**, 1402–1405 (2013).
8. Park, Y. *et al.* Directional ionic transport across the oxide interface enables low-temperature epitaxy of rutile TiO₂. *Nat Commun* **11**, 1401 (2020).
9. Jeong, J. *et al.* Giant reversible, facet-dependent, structural changes in a correlated-electron insulator induced by ionic liquid gating. *Proceedings of the National Academy of Sciences* **112**, 1013–1018 (2015).
10. Nakano, M. *et al.* Collective bulk carrier delocalization driven by electrostatic surface charge accumulation. *Nature* **487**, 459–462 (2012).

REVIEWERS' COMMENTS

Reviewer #2 (Remarks to the Author):

I have thoroughly read the authors' responses and the revised manuscript. The authors have sufficiently addressed my concerns, and the manuscript has been largely improved. I would in particular appreciate the authors for doing the LAO-thickness-dependence experiments, which could in fact support the authors' claim apart from oxygen-vacancy-formation scenario. However, I was disappointed to see that the authors did not include these very important results in the revised manuscript. I would strongly recommend the authors to include those data and relevant discussions in the manuscript. With this further revision, the manuscript would be accepted for publication in Nature Communications.

Reviewer #3 (Remarks to the Author):

Review comments for NCOMMS-23-04444-T

Title: Modulation-Doping a Correlated Electron Insulator

The last revision was well applied with reasonable explanation. It seems that just a few more points are needed for a more sophisticated article. If minor revisions are conducted, I believe it will be accepted soon.

1. All the questions that last reviewers suggested are well established. However, the authors answer only in the rebuttal paper in some cases, not applied in a main manuscript. Please apply all the answers in the main manuscript.

2. The author needs to visualize the TMIT in every R-T curves. Since the TMIT is a kind of indicative that shows the thermal information that can be changed by external condition, the summarized TMIT depending on the experimental condition should be suggested in a form of table or replot form.

Manuscript Number: NCOMMS-23-04444A

Title: Modulation-Doping a Correlated Electron Insulator

Authors: Debasish Mondal¹, Smruti Rekha Mahapatra¹, Abigail M Derrico², Rajeev Kumar Rai³, Jay R Paudel², Christoph Schlueter⁴, Andrei Gloskovskii⁴, Rajdeep Banerjee¹, Atsushi Hariki⁵, Frank M F DeGroot⁶, Dipankar D Sarma¹, Awadhesh Narayan¹, Pavan Nukala³, Alexander X Gray^{2*} and Naga Phani B Aetukuri^{1*}

RESPONSE TO THE REVIEWERS' COMMENTS

We thank all the reviewers for their time and interest in our work. We are pleased to note that all reviewers recommended the publication of our manuscript. Below, we address the reviewers' comments. Reviewers' comments are in blue-colored font and our responses are in black-colored font.

Reviewer #2 (Remarks to the Author):

I have thoroughly read the authors' responses and the revised manuscript. The authors have sufficiently addressed my concerns, and the manuscript has been largely improved. I would in particular appreciate the authors for doing the LAO-thickness-dependence experiments, which could in fact support the authors' claim apart from oxygen-vacancy-formation scenario. However, I was disappointed to see that the authors did not include these very important results in the revised manuscript. I would strongly recommend the authors to include those data and relevant discussions in the manuscript. With this further revision, the manuscript would be accepted for publication in Nature Communications.

We thank the reviewer for their positive assessment of our work and recommendation for publication. The data that the reviewer is suggesting is the LAO thickness-dependent changes to the metal-insulator transition temperature. To address the reviewer's question, we added LAO thickness-dependent data as Fig. 4 in the revised version of the main manuscript. Further, we added the following text in paragraph 2 of page 8 of the main manuscript:

“As further proof of modulation-doping, we performed resistance-temperature measurements on heterostructures with varying LAO spacer layer thicknesses of 2 nm, 4 nm, and 10 nm. The thickness of VO₂ is fixed at 7.5 nm and that of TiO_{2-x} dopant layer is fixed at 3 nm for all three heterostructures. As the thickness of the LAO layer (t_{LAO}) increases, the probability of charge transfer from the TiO_{2-x} dopant layer to the VO₂ layer decreases (Figs. 4a and b). Consistent with this, at the highest t_{LAO} of 10 nm, where the lowest amount of charge transfer is expected from the dopant layer, T_{MIT} is the closest to that of a 7.5 nm VO₂ film with a 2 nm LAO cap layer, but without the dopant layer. Clearly, the dopant layer does not significantly affect the transition temperature when a 10 nm thick LAO spacer layer is used (Fig. 4c). By contrast for the heterostructure with $t_{\text{LAO}} = 2$ nm, the T_{MIT} is shifted by ~ 20 K as discussed earlier. Since the thicknesses of the other layers are fixed, the larger shift in T_{MIT} for the $t_{\text{LAO}} = 2$ nm heterostructure implies an increased charge transfer for the thinner spacer layers. This further reinforces the central conclusion that the shift in the transition temperature is enabled by modulation doping.”

Reviewer #3 (Remarks to the Author):

Review comments for NCOMMS-23-04444-T

Title: Modulation-Doping a Correlated Electron Insulator

The last revision was well applied with reasonable explanation. It seems that just a few more points are needed for a more sophisticated article. If minor revisions are conducted, I believe it will be accepted soon.

We thank the reviewer for their positive assessment of our work and recommendation for publication.

1. All the questions that last reviewers suggested are well established. However, the authors answer only in the rebuttal paper in some cases, not applied in a main manuscript. Please apply all the answers in the main manuscript.

The data that the reviewer is suggesting is the temperature-dependent structural phase transition in VO₂ and LAO thickness-dependent changes to the metal-insulator transition temperature. To address the reviewer's points, we added temperature-dependent structural phase transition data as Fig. 7 in the main manuscript and Supplementary Fig. 18 in Supplementary information. To further include this in the main manuscript we added the following text from paragraph 5 of page 11 to paragraph 2 of page 12 in the revised version of the main text:

“What is unclear however is the presence or absence of the structural phase transition. To probe this, we performed temperature-dependent XRD measurements at 320 K (metallic phase) and 200 K (insulating phase) for a 9.5 nm VO₂ thin film and 9.5 nm and 4.5 nm VO₂ heterostructures. Fig. 7a shows that there is a clear shift in the angular position of the out-of-plane Bragg reflection. This was attributed to the rutile to monoclinic phase transition in previous studies.¹¹ Similar changes to the out-of-plane Bragg reflection were also observed for the 9.5VO₂-het and 4.5VO₂-het across the MIT (Figs. 7b and c). Additionally, the X-ray diffractogram measured at 280 K for the 4.5VO₂-het shows rutile phase characteristics akin to that observed for the diffractogram measured at 320 K (Fig. 7d) suggesting that the electronic phase transition is concomitant with the structural phase transition.

Moreover, in Supplementary Fig. 18, we have also compared the XRD spectra of 9.5VO₂ and 9.5VO₂-het both in the insulating and metallic phases. It is clear from the figure that there is an excellent overlap of the diffractograms for the film and heterostructure, including thickness fringes, for both the structural phases. This is evidence that modulation-doping decreases the phase transition temperature but does not suppress the structural transition. The presence of the structural phase transition even at carrier densities as high as $5 \times 10^{21} \text{ cm}^{-3}$ is suggestive of a strong electron-lattice coupling in VO₂.”

We also added the LAO thickness-dependent changes to the metal-insulator transition temperature data and the associated text to paragraph 2 of page 8 in the revised version of the main manuscript. Also, please see the response to question 1 of reviewer 2.

2. The author needs to visualize the TMIT in every R-T curves. Since the TMIT is a kind of indicative that shows the thermal information that can be changed by external condition, the summarized TMIT depending on the experimental condition should be suggested in a form of table or replot form.

The T_{MIT} values of the R-T curves (for both VO_2 films and heterostructures of different VO_2 thicknesses) are presented in ‘Supplementary Table 1.’ in the Supplementary information file.